# ACCORD: ALLEVIATING CONCEPT COUPLING THROUGH DEPENDENCE REGULARIZATION FOR TEXT-TO-IMAGE DIFFUSION PERSONALIZATION

**Shizhan Liu**[1]**, Hao Zheng**[1]**, Hang Yu**[1*]**, Jianguo Li**[1*]

[1]Ant Group

{chautaulsz, jaruce.zh}@gmail.com, {hyu.hugo, lijg.zero}@antgroup.com

## ABSTRACT

Image personalization enables customizing Text-to-Image models with a few reference images but is plagued by "concept coupling"—the model creating spurious associations between a subject and its context. Existing methods tackle this indirectly, forcing a trade-off between personalization fidelity and text control. This paper is the first to formalize concept coupling as a statistical dependency problem, identifying two root causes: a Denoising Dependence Discrepancy that arises during the generative process, and a Prior Dependence Discrepancy within the learned concept itself. To address this, we introduce ACCORD, a framework with two targeted, plug-and-play regularization losses. The Denoising Decouple Loss minimizes dependency changes across denoising steps, while the Prior Decouple Loss aligns the concept's relational priors with those of its superclass. Extensive experiments across subject, style, and face personalization demonstrate that ACCORD achieves a superior balance between fidelity and text control, consistently improving upon existing methods[1].

## 1 INTRODUCTION

The advancement of Text-to-Image (T2I) Diffusion Models (Ho et al., 2020; Rombach et al., 2022) has lowered the barrier to generating high-quality and imaginative images from text prompts. However, pretrained T2I models often struggle to accurately produce personalized images, such as those depicting private pets or unique artistic styles. As a result, image personalization has gained significant attention, requiring users to provide several reference images related to the personalization target, which enables T2I models to create new images of the target based on text prompts.

The primary challenge of image personalization is "concept coupling". Due to the limited availability and low diversity of reference images for the personalization target (typically 3-6 images often in similar contexts), the model tends to confuse the target with other concepts that appear alongside it in these images. This entanglement hinders the model's ability to accurately control the attributes associated with the personalization target based on text. For example, as shown in Fig. 1, the model may interpret "a person carrying a backpack" as the primary focus, rather than "backpack", because these elements frequently co-occur in the reference images. Consequently, the generated images often deviate from the intended text prompts, frequently including an unintended person in the output.

However, existing methods attempt to mitigate concept coupling through indirect and often heuristic means, fundamentally treating it as a symptom of overfitting rather than addressing its root cause. These approaches, while varied, are ultimately proxies. Open-source approaches fall into four main categories, each with fundamental limitations. Data regularization (Ruiz et al., 2023; Kumari et al., 2023) uses superclass datasets to preserve model priors but risks distorting concept relationships. Weight regularization (Han et al., 2023; Qiu et al., 2023) constrains parameter updates to prevent overfitting, which can indiscriminately degrade fidelity. Loss regularization methods (Qiao et al., 2024; Song et al., 2024) introduce heuristic objectives that lack a direct link to the underlying statistical problem. Region-based methods (Avrahami et al., 2023; Zhang et al., 2024a) are confined

---

*Corresponding authors.

[1]This work was done while Shizhan Liu and Hao Zheng (research intern) were with Ant Group. Code is available at: https://github.com/antgroup/ACCORD

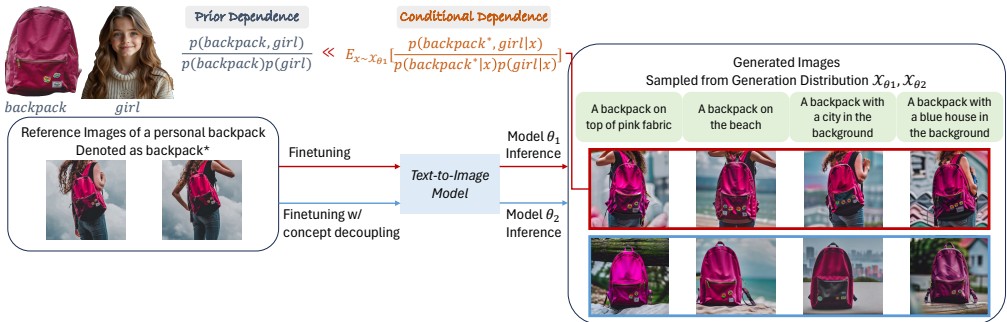

Figure 1: Illustration of the concept coupling problem. The target is a "backpack*", but reference images always pair it with a "girl". Standard finetuning incorrectly learns to bind these concepts, causing the model to generate the unwanted 'girl' and violate the text prompt.

to spatially separable objects and fail for global attributes like style. In addition, even powerful closed-source models like GPT-4o exhibit inconsistencies and artifacts stemming from this issue, as observed in recent empirical studies (Chen et al., 2025; Yan et al., 2025). By focusing on symptoms like parameter drift or feature entanglement, these approaches fail to directly model and minimize the unintended statistical dependencies that define concept coupling, leaving a critical gap for a more principled solution.

In this paper, we fill this gap by proposing a new paradigm: we are the first to formally frame concept coupling as a tractable statistical dependency problem. Our analysis reveals that this unwanted dependency originates from two distinct and measurable sources: a **Denoising Dependence Discrepancy** introduced during the generative process, and a **Prior Dependence Discrepancy** inherent in the learned personalized concept. This new formalism moves beyond heuristic fixes and allows us to directly diagnose and treat the problem at its core.

To operationalize this insight, we introduce ACCORD (**A**lleviating **C**oncept **CO**upling th**R**ough **D**ependence regularization), a plug-and-play framework with two targeted, theoretically-grounded regularization losses. The **Denoising Decouple Loss (DDLoss)** directly minimizes the dependency discrepancy that accumulates during the denoising process by leveraging the diffusion model as an implicit classifier. Complementing this, the **Prior Decouple Loss (PDLoss)** corrects the prior dependency of the learned concept by aligning its relationship with other concepts to that of its superclass in CLIP's semantic space. Together, these losses enable ACCORD to directly minimize concept coupling without relying on regularization datasets or overly restrictive weight constraints. Experiments demonstrate that the proposed loss functions alleviate the concept coupling issue in image personalization more effectively, achieving a better balance between text control and personalization fidelity. Our contributions can be summarized as follows:

- We are among the first to formally **formulate concept coupling in image personalization as a statistical problem of unintended dependencies** and propose ACCORD, a **plug-and-play** method that directly addresses concept coupling without requiring regularization datasets or extensive weight constraints.

- We **identify two distinct sources of dependence discrepancies in concept coupling**: Denoising Dependence Discrepancy and Prior Dependence Discrepancy. To address these discrepancies, we propose Denoising Decouple Loss and Prior Decouple Loss, respectively.

- Experimental results demonstrate the superiority of ACCORD in image personalization. Moreover, the proposed losses **prove effective in zero-shot conditional control tasks**, highlighting the general applicability of our decoupling principle beyond test-time finetuning.

## 2 RELATED WORKS

**Test-Time Finetuning-based Image Personalization**: Test-time fine-tuning, on which this paper mainly focuses, adapts pre-trained T2I models to reference images, offering flexible and balanced personalization at the cost of time and computation.

Existing test-time fine-tuning methods attempt to mitigate concept coupling through indirect means, which can be grouped into four main categories of proxy-based regularization, all of which treat the symptoms of the problem rather than its root cause: **Data regularization** (Ruiz et al., 2023; Kumari et al., 2023) augments training with images of both the personalization target and its superclass.

While intended to prevent overfitting, this approach is a blunt instrument; limited regularization dataset size and distribution gaps can hinder accurate modeling of concept relationships and reduce personalization fidelity. Although (He et al., 2025) use LLMs to design structured prompts for diverse regularization data to improve regularization effectiveness, this introduces LLM-induced concept dependencies that may not reflect their true prior relationships. **Weight regularization** methods (Gal et al., 2022; Hu et al., 2021; Han et al., 2023; Qiu et al., 2023; Chen et al., 2024a) constrain parameter updates to prevent overfitting. For example, PaRa constrains the parameter space by reducing the dimensionality of the output matrix, thereby preventing overfitting. Yet, weight regularization can also diminish fidelity by indiscriminately restricting the model's capacity to learn target-specific details. **Loss regularization** approaches, like MagiCapture (Hyung et al., 2023) and Facechain-SuDe (Qiao et al., 2024), introduce objectives such as masked reconstruction or superclass inheritance to promote decoupling. However, their reliance on empirically chosen objectives means they lack a formal basis for why these heuristics should reduce the statistical dependencies at the core of concept coupling. **Region regularization** limit subjects to specific regions in the attention map (Avrahami et al., 2023; Zhang et al., 2024a; Hao et al., 2024) or alternatively refines subject generation by constraining the cross attention map, as in Attend-and-Excite (Chefer et al., 2023). But this spatial proxy for conceptual separation is limited to spatially distinct subjects and struggles with global concepts like style or viewpoint. Perfusion (Tewel et al., 2023) further combines weight regularization and region regularization using gated rank-1 updates and key-locking. However, it still cannot theoretically constrain the statistical dependencies between concepts.

Unlike these proxy-based strategies that indirectly target symptoms like overfitting, our work is the first to directly model concept coupling as an excessive inter-concept dependency. We then introduce two targeted, dependency-regularization loss functions to principledly minimize it.

**Zero-shot Image Personalization**: Unlike test-time finetuning, zero-shot image personalization avoids test-time training but relies heavily on large-scale pretraining data. While recent closed-source models (e.g., GPT4o, Gemini 2.0) outperform open-source ones in zero-shot personalization (Wang et al., 2024c; Xiao et al., 2025), they still face issues such as inconsistencies (Yan et al., 2025) and copy-paste artifacts (Chen et al., 2025). Most open-source models are limited to specific domains (e.g., faces, objects) and cannot fully address diverse personalization needs. Representative approaches include: for **subject personalization**, methods like InstantBooth (Shi et al., 2024), BLIP-Diffusion (Li et al., 2024), and ELITE (Wei et al., 2023) focus on improved visual encoding and hierarchical concept mapping, while others (Song et al., 2024) tackle weak text control by removing the projection of visual embeddings onto text embeddings. For **face personalization**, InstantID (Wang et al., 2024b) extracts both appearance and structural features from cropped faces. For **style personalization**, InstantStyle(Wang et al., 2024a) performs style transfer by injecting IP-Adapter (Ye et al., 2023) features into style-related layers of SDXL (Podell et al., 2023).

While this paper places less emphasis on zero-shot image personalization, **our experiments demonstrate the potential applicability of ACCORD to these approaches.**

## 3 METHOD

### 3.1 TEXT-TO-IMAGE (T2I) DIFFUSION MODELS

We begin with a brief introduction to the T2I Diffusion Model (Ho et al., 2020), which establishes a mapping between the image distribution and the standard Gaussian distribution via a forward noise-adding process and a reverse denoising process. Specifically, the forward process is composed of $T$ steps, gradually introducing Gaussian noise into a clear image or its latent code $\mathbf{x}_0$. The noisy code at time step $t \in \{1, 2, ..., T\}$ is calculated as follows:

$$\mathbf{x}_t = \sqrt{\alpha_t}\mathbf{x}_0 + \sqrt{1 - \alpha_t}\boldsymbol{\epsilon}, \tag{1}$$

where $\boldsymbol{\epsilon} \sim \mathcal{N}(\mathbf{0}, \boldsymbol{I})$ represents Gaussian noise, and $\alpha_t$ modulates the retention of the original image, decreasing as $t$ increases. When $T$ is sufficiently large, $\mathbf{x}_T$ is approximately a standard Gaussian.

The reverse process is modeled as a Markov chain, where a network $\mathcal{U}_\theta$ with parameters $\theta$ is used to estimate the parameters of the true posterior distribution $q(\mathbf{x}_{t-1}|\mathbf{x}_t, \mathbf{x}_0)$ based on $t$ and $\mathbf{x}_t$, thereby denoising the noisy code. The optimization objective can be expressed as:

$$\mathbb{E}_{\mathbf{x}_0, \boldsymbol{\epsilon}, \mathbf{c}, t}[\frac{1}{2\boldsymbol{\sigma}_t^2}\|\mathbf{x}_{t-1} - \mathcal{U}_\theta(\mathbf{x}_t, \mathbf{c}, t)\|^2], \tag{2}$$

where $\boldsymbol{\sigma}_t$ represents the standard deviation of the noisy code at time step $t$, and $\mathcal{U}_\theta(\mathbf{x}_t, \mathbf{c}, t)$ is the output of the denoising model. During inference, the noisy code $\mathbf{x}_{t-1}$ at time step $t-1$ can be sampled from $\mathcal{N}(\mathcal{U}_\theta(\mathbf{x}_t, \mathbf{c}, t), \boldsymbol{\sigma}_t^2 \boldsymbol{I})$, yielding $\mathbf{x}_{t-1} = \mathcal{U}_\theta(\mathbf{x}_t, \mathbf{c}, t) + \boldsymbol{\sigma}_t \boldsymbol{\epsilon}_t$, where $\boldsymbol{\epsilon} \sim \mathcal{N}(\mathbf{0}, \boldsymbol{I})$. Note that the text representation or the conditioning information $\mathbf{c}$ is also fed into the denoising model to control the generation.

To facilitate subsequent discussions, we further introduce the **conditional dependence coefficient** $r$ for two concepts $\mathbf{c}_p$ and $\mathbf{c}_g$, given the model's denoised output based on $(\mathbf{c}_p, \mathbf{c}_g)$ at time step $t$, i.e., $\mathbf{x}_{\theta,t} := \mathcal{U}_\theta(\mathbf{x}_{t+1}, (\mathbf{c}_p, \mathbf{c}_g), t+1)$. This coefficient can be defined as the ratio between the joint probability of the two concepts occurring together in $\mathbf{x}_{\theta,t}$ and the probability of their independent occurrences in the same representation:

$$r(\mathbf{c}_p, \mathbf{c}_g | \mathbf{x}_{\theta,t}) = \frac{p(\mathbf{c}_p, \mathbf{c}_g | \mathbf{x}_{\theta,t})}{p(\mathbf{c}_p | \mathbf{x}_{\theta,t}) p(\mathbf{c}_g | \mathbf{x}_{\theta,t})}. \tag{3}$$

According to probability theory, $\mathbf{c}_p$ and $\mathbf{c}_g$ are conditionally independent given $\mathbf{x}_{\theta,t}$ when $r(\mathbf{c}_p, \mathbf{c}_g | \mathbf{x}_{\theta,t}) = 1$; they are conditionally dependent otherwise.

We provide a notation summary in *Tab. 7* in the Appendix.

## 3.2 CONCEPT COUPLING IN IMAGE PERSONALIZATION

Test-time finetuning methods are designed to achieve image personalization by fine-tuning a pre-trained T2I model on a limited set of reference images with the personalization target, denoted as $\mathbb{D} = \{(\mathbf{x}^i, \mathbf{c}^i)\}_{i=1}^N$. Here, $N$ is the number of training samples. $\mathbf{x}^i$ and $\mathbf{c}^i$ represent the reference image and the corresponding generation condition for the $i$-th pair, respectively. Note that $\mathbf{c}^i$ can be either an image caption or a combination of the caption and visual features extracted from the reference images for personalization purposes. In instances where captions for $\mathbf{x}^i$ are absent, we employ Vision Language Models (VLMs) (Chen et al., 2024b) to generate image captions, aligning with practices in the community. This approach, compared to using prompt templates (Ruiz et al., 2023), yields more meaningful textual concepts and assists in the decoupling of concepts.

One issue that plagues image personalization is concept coupling. As illustrated in Fig. 1, although the personalization target $\mathbf{c}_p$ is a specifically designed red backpack, the training set $\mathbb{D}$ consistently pairs the personalized backpack $\mathbf{c}_p$ with a girl $\mathbf{c}_g$. Consequently, the adapted T2I model often tends to generate an additional girl during inference, which contradicts the original prompt. This phenomenon can be statistically characterized as:

$$\mathbb{E}_{\mathbf{x}_\theta}[|\log r(\mathbf{c}_p, \mathbf{c}_g | \mathbf{x}_{\theta,0}) - \log r(\mathbf{c}_s, \mathbf{c}_g)|] \gg 0, \tag{4}$$

where $|\cdot|$ denotes the absolute value, $\mathbf{x}_{\theta,0}$ denotes the image generated by the T2I model or its latent code, $\mathbf{c}_p$ and $\mathbf{c}_g$ represent the **p**ersonalized target condition and the **g**eneral text condition respectively. The **p**ersonalization target condition $\mathbf{c}_p$ can be either the textual trigger words used during LoRA training, the text embedding from (Gal et al., 2022), or the image representation from (Ye et al., 2023), while $\mathbf{c}_s$ denotes **s**uperclass of $\mathbf{c}_p$. Additionally, $r(\mathbf{c}_s, \mathbf{c}_g) = p(\mathbf{c}_s, \mathbf{c}_g)/p(\mathbf{c}_s)/p(\mathbf{c}_g)$. In this context, $\mathbf{c}_s$ embodies a general backpack, thus encompassing the overall properties of $\mathbf{c}_p$ and further characterizing the inherent relationships with other general concepts represented by $\mathbf{c}_g$ (Ruiz et al., 2023; Qiao et al., 2024). The essence of the equation above is that the generated images $\mathbf{x}_{\theta,0}$ typically introduce additional interdependencies between $\mathbf{c}_p$ and $\mathbf{c}_g$ that are not present in the inherent prior relationships between $\mathbf{c}_s$ and $\mathbf{c}_g$. Indeed,

**Lemma 1.** $\mathbb{E}_{\mathbf{x}_\theta}[|\log r(\mathbf{c}_p, \mathbf{c}_g | \mathbf{x}_{\theta,0}) - \log r(\mathbf{c}_s, \mathbf{c}_g)|] > 0$ *holds when either (i)* $r(\mathbf{c}_p, \mathbf{c}_g | \mathbf{x}_{\theta,0}) > r(\mathbf{c}_s, \mathbf{c}_g)$ *(overly positive dependence) or (ii)* $r(\mathbf{c}_p, \mathbf{c}_g | \mathbf{x}_{\theta,0}) < r(\mathbf{c}_s, \mathbf{c}_g)$ *(overly negative dependence). The equality is achieved if and only if* $r(\mathbf{c}_p, \mathbf{c}_g | \mathbf{x}_{\theta,0}) = r(\mathbf{c}_s, \mathbf{c}_g)$.

Thus, the fundamental goal of concept decoupling is to correct the conditional dependence coefficient between $\mathbf{c}_p$ and $\mathbf{c}_g$ in the generated images so that it approximates the prior concept dependence between $\mathbf{c}_s$ and $\mathbf{c}_g$.

## 3.3 SOURCES OF DEPENDENCE DISCREPANCIES

The direct computation and minimization of the left-hand side (LHS) of Eq. (4) pose significant challenges due to the absence of a closed-form expression. Instead, we analyze this discrepancy by introducing an intermediate term $\log r(\mathbf{c}_p, \mathbf{c}_g | \mathbf{x}_T)$, which allows us to separate the total discrepancy into two meaningful and computable components, as formalized in Theorem 1.

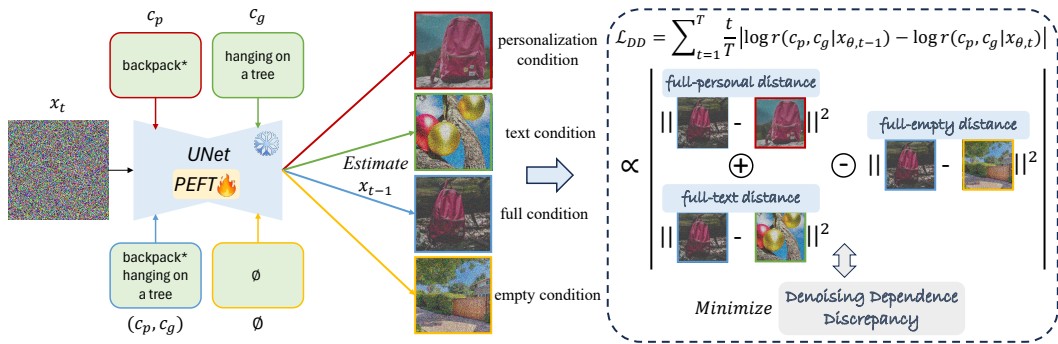

Figure 2: Denoising Decouple Loss $\mathcal{L}_{\mathrm{DD}}$. The UNet estimates $\mathbf{x}_{t-1}$ based on $\mathbf{x}_t$ and four different conditions, then constrains the relationships between the four denoising results. The objective of $\mathcal{L}_{\mathrm{DD}}$ is to prevent the conditional dependence coefficient between the personalization target $\mathbf{c}_p$ and the general text condition $\mathbf{c}_g$ from varying significantly between adjacent timesteps.

**Theorem 1.** *The LHS of Eq. (4) can be decomposed into the following two terms:*

$$\mathbb{E}_{\mathbf{x}_\theta}\Big[\big|\underbrace{\log r(\mathbf{c}_p,\mathbf{c}_g|\mathbf{x}_{\theta,0}) - \log r(\mathbf{c}_p,\mathbf{c}_g|\mathbf{x}_T)}_{\text{① Denoising Dependence Discrepancy}} + \underbrace{\log r(\mathbf{c}_p,\mathbf{c}_g) - \log r(\mathbf{c}_s,\mathbf{c}_g)}_{\text{② Prior Dependence Discrepancy}}\big|\Big], \qquad (5)$$

*where $\mathbf{x}_T$ denotes multivariate standard Gaussian noise.*

Since $\mathbf{x}_T$ is Gaussian noise sampled independently of the conditions $\mathbf{c}_p$ and $\mathbf{c}_g$, it follows that $\log r(\mathbf{c}_p,\mathbf{c}_g|\mathbf{x}_T) = \log r(\mathbf{c}_p,\mathbf{c}_g)$. The detailed proof is provided in Appendix A. Therefore, the expression in (5) equals the left-hand side of Eq. (4).

The **denoising dependence discrepancy** ① captures the change in conditional dependence between $\mathbf{c}_p$ and $\mathbf{c}_g$ introduced during denoising, whereas the **prior dependence discrepancy** ② reflects the alteration in prior dependence due to deviations of $\mathbf{c}_p$ from $\mathbf{c}_s$. The conditional dependence coefficient of $\mathbf{c}_p$ and $\mathbf{c}_g$ on $\mathbf{x}_T$, $\log r(\mathbf{c}_p,\mathbf{c}_g)$, bridges the denoising dependence and prior dependence.

Building on this decomposition, we propose **ACCORD**, a plug-and-play method comprising two loss functions: the **Denoising Decouple Loss (DDLoss)** and the **Prior Decouple Loss (PDLoss)**. The DDLoss minimizes the denoising dependence discrepancy by leveraging the implicit classification capabilities of the diffusion model, while the PDLoss alleviates prior dependence discrepancy, particularly when $\mathbf{c}_p$ is trainable, by utilizing the classification capability of CLIP. Collectively, these strategies work synergistically to minimize concept coupling, which will be elaborated below.

### 3.4 DENOISING DECOUPLE LOSS (DDLOSS)

We first elaborate on the DDloss, which specifically targets the denoising dependence discrepancy. Directly minimizing the denoising dependence discrepancy term in Eq. (5) is not well-aligned with the time step sampling mechanism employed during the training of diffusion models. This incompatibility arises because the term connects the first and last time steps, bypassing the relationships between successive steps. To address this issue, we propose to relax this term by upper-bounding it with the sum of dependence discrepancies between adjacent denoising steps:

$$|\log r(\mathbf{c}_p,\mathbf{c}_g|\mathbf{x}_{\theta,0}) - \log r(\mathbf{c}_p,\mathbf{c}_g|\mathbf{x}_T)| = |\sum_{t=1}^{T} \log r(\mathbf{c}_p,\mathbf{c}_g|\mathbf{x}_{\theta,t-1}) - \log r(\mathbf{c}_p,\mathbf{c}_g|\mathbf{x}_{\theta,t})|$$

$$\leq \sum_{t=1}^{T} |\log r(\mathbf{c}_p,\mathbf{c}_g|\mathbf{x}_{\theta,t-1}) - \log r(\mathbf{c}_p,\mathbf{c}_g|\mathbf{x}_{\theta,t})|. \qquad (6)$$

This relaxation follows from the triangle inequality. Minimizing this upper bound effectively discourages the conditional dependence between the personalization target and any other concepts from changing abruptly between consecutive denoising steps.

Next, by exploiting the diffusion model as an implicit classifier (Qiao et al., 2024), we can derive a closed-form expression for $\log r(\mathbf{c}_p,\mathbf{c}_g|\mathbf{x}_{\theta,t-1}) - \log r(\mathbf{c}_p,\mathbf{c}_g|\mathbf{x}_{\theta,t})$:

**Theorem 2.** *The dependence discrepancy between successive time steps in diffusion models can be computed as:*

$$\log r(\mathbf{c}_p,\mathbf{c}_g|\mathbf{x}_{\theta,t-1}) - \log r(\mathbf{c}_p,\mathbf{c}_g|\mathbf{x}_{\theta,t})$$

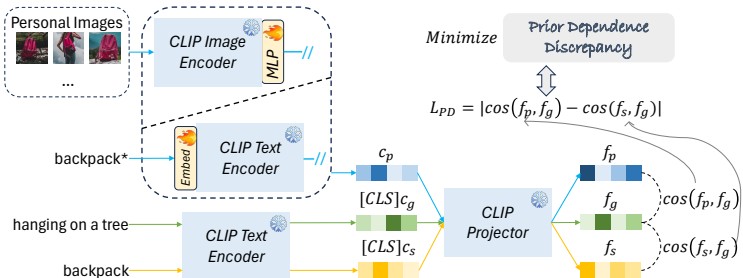

Figure 3: Prior Decouple Loss $\mathcal{L}_{\text{PD}}$. Either the Image Encoder or the Text Encoder of CLIP can be used to generate $\mathbf{c}_p$. The purpose of $\mathcal{L}_{\text{PD}}$ is to prevent excessive prior dependence between $\mathbf{c}_p$ and the general text condition $\mathbf{c}_g$. We first use the CLIP projector to map $\mathbf{c}_p$ and $\mathbf{c}_g$ into $\mathbf{f}_s$ and $\mathbf{f}_g$, respectively, and then minimize the absolute difference between $\cos(\mathbf{f}_p, \mathbf{f}_g)$ and $\cos(\mathbf{f}_s, \mathbf{f}_g)$.

$$
= \frac{1}{2\boldsymbol{\sigma}_t^2} \Big[ \|\mathcal{U}_\theta\big(\mathbf{x}_t, (\mathbf{c}_p, \mathbf{c}_g), t\big) - \mathcal{U}_\theta\big(\mathbf{x}_{\theta,t}, \mathbf{c}_p, t\big)\|^2 + \|\mathcal{U}_\theta\big(\mathbf{x}_t, (\mathbf{c}_p, \mathbf{c}_g), t\big) - \mathcal{U}_\theta\big(\mathbf{x}_{\theta,t}, \mathbf{c}_g, t\big)\|^2
$$
$$
- \|\mathcal{U}_\theta\big(\mathbf{x}_t, (\mathbf{c}_p, \mathbf{c}_g), t\big) - \mathcal{U}_\theta\big(\mathbf{x}_{\theta,t}, \varnothing, t\big)\|^2 \Big], \tag{7}
$$

*where $\varnothing$ denotes an empty control condition.*

Theorem 2 follows from Bayes' theorem and the Gaussianity of noisy latents at timestep $t-1$; see *Appendix B* for details. Intuitively, Eq. (7) measures dependence changes by comparing the model's prediction for the joint concept $(\mathbf{c}_p, \mathbf{c}_g)$ against its predictions for each individual concept and the empty condition, thus penalizing deviations that imply a change in their relationship. Finally, we define the DDLoss as:

$$
\mathcal{L}_{\text{DD}} = \sum_{t=1}^{T} \frac{t}{T} |\log r(\mathbf{c}_p, \mathbf{c}_g | \mathbf{x}_{\theta, t-1}) - \log r(\mathbf{c}_p, \mathbf{c}_g | \mathbf{x}_{\theta, t})|. \tag{8}
$$

In this formulation, $\mathcal{L}_{\text{DD}}^t$ with a larger $t$ contributes more to concept decoupling due to loss accumulation. Therefore, we scale $\mathcal{L}_{\text{DD}}^t$ by a linearly time-varying weight $t/T$. Moreover, to compute the DDLoss in practice, we use $\mathbf{x}_t$ instead of $\mathbf{x}_{\theta,t}$. This approximation is effective for two reasons: (i) During diffusion training, we sample individual time steps using Eq. (1) rather than iterating from time step $T$ to 0. Consequently, $\mathbf{x}_{\theta,t}$ is not directly accessible when denoising from $t$ to $t-1$. (ii) $\mathbf{x}_t$ serves as an unbiased estimate of $\mathbf{x}_{\theta,t}$. Additionally, we stop the gradients for $\mathcal{U}_\theta(\mathbf{x}_t, \mathbf{c}_g, t)$ and $\mathcal{U}_\theta(\mathbf{x}_t, \varnothing, t)$, following Facechain-SuDe (Qiao et al., 2024), to prevent damaging the model's prior knowledge. For ease of understanding, we show the computation of DDLoss in Fig. 2.

## 3.5 PRIOR DECOUPLE LOSS (PDLOSS)

When $\mathbf{c}_p$ remains fixed and close to $\mathbf{c}_s$ during training, the coupling of concepts primarily arises from the first term in Eq. (5), specifically the denoising dependence discrepancy. In this context, minimizing only the DDLoss allows the personalized target to retain its superclass's relationship with various text control conditions. However, it is worth noting that $\mathbf{c}_p$ can also be trained as either the CLIP text representation (Gal et al., 2022) or the representation extracted from reference images by the CLIP image encoder (Ye et al., 2023), to better capture the details of the personalization target. Yet, it is crucial to note that training $\mathbf{c}_p$ may cause $\mathbf{c}_p$ to diverge from $\mathbf{c}_s$ and so drastically increase the prior dependence discrepancy (see ② in (5)). As a remedy, we introduce the PDLoss. Specifically, the prior dependence discrepancy can be equivalently written as:

$$
\log r(\mathbf{c}_p, \mathbf{c}_g) - \log r(\mathbf{c}_s, \mathbf{c}_g) = \log \frac{p(\mathbf{c}_g | \mathbf{c}_p)}{p(\mathbf{c}_g | \mathbf{c}_s)}. \tag{9}
$$

This equation shows that reducing prior dependence discrepancy involves aligning the conditional probabilities $p(\mathbf{c}_g | \mathbf{c}_p)$ and $p(\mathbf{c}_g | \mathbf{c}_s)$. Unfortunately, the diffusion model does not facilitate this alignment because Eq. (9) is independent of the denoising process. Therefore, we leverage the semantic space of CLIP, which is inherently density-aligned due to its training objective. Specifically, CLIP is trained with the InfoNCE loss, whose optimization objective is to estimate a density ratio relative to the noise, as shown in Lemma 2 (equivalent to Eq. (2) in InfoNCE (Oord et al., 2018)).

Table 1: Quantitative results on DreamBench. The "*" indicates results using per-subject/style loss weights, tuned on a small validation set. "Params." indicates the number of tunable parameters. The W(in)/L(oss) rate is calculated by pairwise human comparison between the anonymous generated results of the baseline and Ours*, with ties omitted. 'PA' denotes percent agreement, namely the percentage of samples receiving consistent judgments from human annotators. The comparison methods improved based on the baseline are *italicized*.

| Method | CLIP-T↑ | BLIP-T↑ | CLIP-I↑ | DINO-I↑ | W↑/L↓ (%) | PA (%) | Params. |
|---|---|---|---|---|---|---|---|
| DreamBooth (DB) | 30.3 | 40.3 | 74.0 | 69.3 | 18.1/**75.7** | 73.3 | 819.7 M |
| *CoRe-SD1.5* | 29.4 | 40.3 | 78.3 | 72.3 | 19.2/**61.7** | 60.0 | 819.7M |
| *Facechain-SuDe* | **31.4** | 41.6 | 74.3 | 70.5 | 14.2/**69.2** | 70.0 | 819.7 M |
| DB w/ Ours | 31.1 (+0.8) | 42.1 (+1.8) | 77.8 (+3.8) | 73.5 (+4.2) | -/- | - | 819.7 M |
| **DB w/ Ours\*** | 31.3 (+1.0) | **42.1** (+1.8) | **78.6** (+4.6) | **74.4** (+5.1) | -/- | - | 819.7 M |
| CustomDiffusion (CD) | 34.2 | 45.4 | 62.7 | 56.9 | 8.1/**88.1** | 76.7 | 18.3 M |
| *ClassDiffusion* | **34.3** | 45.8 | 61.3 | 55.0 | 7.5/**75.8** | 80.0 | 18.3M |
| CD w/ Ours | 33.9 (-0.3) | 46.4 (+1.0) | 71.1 (+8.4) | 65.2 (+8.3) | -/- | - | 18.3 M |
| **CD w/ Ours\*** | 34.1 (-0.1) | **46.6** (+1.2) | **71.4** (+8.7) | **65.6** (+8.7) | -/- | - | 18.3 M |
| LoRA (SDXL) | 34.5 | 47.0 | 76.3 | 72.1 | 17.6/**70.5** | 70.0 | 92.9 M |
| *SVDiff* | 32.7 | 43.7 | 72.6 | 66.6 | 1.7/**85.0** | 83.3 | 0.2 M |
| Omnigen | **35.3** | 47.8 | 73.9 | 68.6 | 30.8/**48.3** | 46.7 | 3.8 B |
| LoRA w/ Ours | 35.1 (+0.6) | **47.8** (+0.8) | 76.8 (+0.5) | 71.9 (-0.2) | -/- | - | 92.9 M |
| **LoRA w/ Ours\*** | 35.2 (+0.7) | 47.7 (+0.7) | **77.1** (+0.8) | **72.4** (+0.3) | -/- | - | 92.9 M |
| VisualEncoder (VE) | 25.9 | 36.1 | 79.1 | 75.5 | 21.1/**67.6** | 56.7 | 3.0 M |
| VE w/ Ours | 25.9 (+0.0) | 35.8 (-0.3) | 80.0 (+0.9) | 76.0 (+0.5) | -/- | - | 3.0 M |
| **VE w/ Ours\*** | **26.3** (+0.4) | **36.1** (+0.0) | **80.4** (+1.3) | **76.7** (+1.2) | -/- | - | 3.0 M |

**Lemma 2.** *For an observation $\mathbf{c}_j$ and condition $\mathbf{c}_k$, the InfoNCE objective seeks to estimate a function $\mathcal{F}(\mathbf{c}_j, \mathbf{c}_k)$ which is proportional to the following density ratio: $\mathcal{F}(\mathbf{c}_j, \mathbf{c}_k) \propto \frac{p(\mathbf{c}_j|\mathbf{c}_k)}{p(\mathbf{c}_j)}$.*

In the case of CLIP, we denote $\tau$ as the temperature coefficient, and let $\mathbf{f}_j$ and $\mathbf{f}_k$ be the projected features of two concepts $\mathbf{c}_j$ and $\mathbf{c}_k$ using the CLIP projection head. Then the function $\mathcal{F}(\mathbf{c}_j, \mathbf{c}_k)$ is instantiated as the scaled cosine similarity $\tau \cos(\mathbf{f}_j, \mathbf{f}_k)$ in the joint embedding space. Thus, we have the following approximation:

$$\tau \cos(\mathbf{f}_j, \mathbf{f}_k) \propto \frac{p(\mathbf{c}_j|\mathbf{c}_k)}{p(\mathbf{c}_j)}. \tag{10}$$

We then align $p(\mathbf{c}_g|\mathbf{c}_p)$ and $p(\mathbf{c}_g|\mathbf{c}_s)$ by ensuring that $\cos(\mathbf{f}_p, \mathbf{f}_g)$ and $\cos(\mathbf{f}_s, \mathbf{f}_g)$ are closely matched.

**Theorem 3.** *The prior dependence discrepancy can be minimized by the following PDLoss:*

$$\mathcal{L}_{PD} = \mathbb{E}_{\mathbf{c}_g}[|\cos(\mathbf{f}_p, \mathbf{f}_g) - \cos(\mathbf{f}_s, \mathbf{f}_g)|] \tag{11}$$

$$\propto \mathbb{E}_{\mathbf{c}_g}\left[\left|\frac{p(\mathbf{c}_g|\mathbf{c}_p) - p(\mathbf{c}_g|\mathbf{c}_s)}{p(\mathbf{c}_g)}\right|\right]. \tag{12}$$

The denominator $p(\mathbf{c}_g)$ in Eq. (12) is not optimizable. Thus, minimizing PDLoss encourages minimization of $|p(\mathbf{c}_g|\mathbf{c}_p) - p(\mathbf{c}_g|\mathbf{c}_s)|$, namely aligns $p(\mathbf{c}_g|\mathbf{c}_p)$ and $p(\mathbf{c}_g|\mathbf{c}_s)$. To facilitate understanding, we show the computation diagram of PDLoss in Fig. 3. Note that in our formulation, $\mathbf{c}_s$ is the text embedding of the superclass (e.g., backpack) given by the CLIP Text Encoder, while $\mathbf{c}_p$ (e.g., the specifically designed red backpack in Fig. 1) is often set as either a trainable text embedding in CLIP or a visual representation mapped to the same space. As both $\mathbf{c}_s$ and $\mathbf{c}_p$ exist in this shared space, they fulfill the necessary conditions to apply Eq. (10). We empirically validate this design choice against several alternative objectives in Appendix D, demonstrating that our formulation provides the best balance between text control and personalization fidelity.

In summary, our framework is both modular and broadly applicable. DDLoss can be applied to any fine-tuning-based personalization method without architectural changes, while PDLoss further benefits scenarios where the personalized embedding $\mathbf{c}_p$ is trainable. Depending on the personalization setup, the two losses can be used independently or together, making ACCORD a flexible plug-and-play regularizer for alleviating concept coupling.

## 4 EXPERIMENTS

**Experimental Setup.** We evaluate our method on diverse image personalization tasks: subject-driven personalization using DreamBench (Ruiz et al., 2023), style personalization with StyleBench (Junyao et al., 2024), and zero-shot face personalization on FFHQ (Karras et al., 2021).

For subject personalization, we use CLIP-T (Ruiz et al., 2023) and BLIP2-T (Qiao et al., 2024) for text alignment, and CLIP-I and DINO-I (Ruiz et al., 2023) for subject fidelity[2]. To reduce background interference, subjects in both real and generated images are segmented using the Reference Segmentation Model (Zhang et al., 2024b). For style personalization, CLIP-T and BLIP-T measure prompt-image alignment, while style similarity is computed using the mean Gram matrix distance (Gram-D) (Gatys et al., 2016). For face personalization, besides CLIP-T and BLIP-T, we further assess facial similarity using Face-Sim (the average cosine similarity of ArcFace (Deng et al., 2019) embeddings for real and generated faces), validated by IP-Adapter (Ye et al., 2023). We compare our approach with 10 baselines (Hu et al., 2021; Ruiz et al., 2023; Kumari et al., 2023; Han et al., 2023; Ye et al., 2023; Qiao et al., 2024; Huang et al., 2025; Wu et al., 2025; Frenkel et al., 2024; Xiao et al., 2025). Our losses are integrated as a plug-and-play module, leaving architectures and hyperparameters unchanged. Only DDLoss is used for methods that do not update the personalized embedding (e.g., DreamBooth, LoRA), while both losses are applied otherwise.

## 4.1 PERSONALIZATION EXPERIMENTS

We report quantitative results for subject, style, and face personalization in Tabs. 1-5, and visualization results in Figs. 4-5. More visualizations are provided in *Appendix K*.

**Subject Personalization.** We compare the performance of different methods on subject personalization in Tab. 1 and Fig. 4. Compared methods include: data-regularization-based Dreambooth and CustomDiffusion, weight-regularization-based LoRA and SVDiff, loss-regularization-based Facechain-SuDe and ClassDiffusion, region-regularization-based CoRe, and the zero-shot method Omnigen. It can be observed that: (i) Our method improves Dream-

Table 2: Quantitative results on StyleBench. The "*" denotes adjusting DDLoss and PDLoss weights across different styles. "Gram-D" is the gram matrix distance.

| Method | CLIP-T↑ | BLIP-T↑ | Gram-D↓ |
|---|---|---|---|
| DreamBooth | 31.3 | 46.6 | 42728 |
| *Facechain-SuDe* | 31.0 | 45.8 | **39978** |
| DB w/ Ours | 31.9 (+0.6) | **47.3** (+0.7) | 42524 (-0.5%) |
| DB w/ Ours* | **32.0** (+0.7) | 47.2 (+0.6) | 41911 (-1.9%) |
| CustomDiffusion | 31.2 | 47.7 | 53347 |
| *ClassDiffusion* | 31.8 | 48.4 | 52998 |
| CD w/ Ours | 31.7 (+0.5) | 48.5 (+0.8) | 48649 (-8.8%) |
| CD w/ Ours* | **31.8** (+0.6) | **48.5** (+0.8) | **47852** (-10.3%) |
| LoRA (SDXL) | 33.1 | 49.7 | 47193 |
| *Omnigen* | 31.9 | 47.5 | 45067 |
| *B-LoRA* | 33.0 | 49.0 | **42048** |
| LoRA (SDXL) w/ Ours | 33.6 (+0.5) | 50.7 (+1.0) | 47693 (+1.1%) |
| LoRA (SDXL) w/ Ours* | **33.6** (+0.5) | **50.7** (+1.0) | 46361 (-1.8%) |
| VisualEncoder | 17.7 | 30.2 | 32176 |
| VE w/ Ours | 17.7 (+0.0) | 30.3 (+0.1) | 31382 (-2.5%) |
| VE w/ Ours* | **18.4** (+0.7) | **30.9** (+0.7) | **27984** (-13.0%) |

Booth and CustomDiffusion by a large margin. They utilize a regularization dataset to enhance text alignment, but may inadvertently sacrifice subject fidelity. This issue arises because the regularization dataset may confuse the model in distinguishing which concepts from the reference images require personalization and which do not. As a result, the model's focus on the personalization target is diminished, leading to a loss of personalization fidelity. **Our method significantly improves personalization fidelity by complementing the regularization dataset with explicit concept decoupling.** (ii) When compared to LoRA and VisualEncoder, which do not utilize a regularization dataset, ACCORD shows smaller improvements. Nevertheless, **ACCORD is able to enhance both text alignment and subject fidelity simultaneously, while most existing image personalization methods** (Han et al., 2023; Qiao et al., 2024; Wu et al., 2025) **tend to improve one aspect at the expense of the other**. Notably, LoRA (SDXL) with ACCORD even outperforms the powerful Omnigen with 3.8B parameters, a testament to the efficiency and effectiveness of our approach. (iii) Our DDLoss and PDLoss significantly enhance the performance of existing baselines in a **plug-and-play** manner. Compared to the similar plug-and-play loss regularization methods Facechain-SuDe, ClassDiffusion and CoRe, our proposed loss functions offer stronger regularization by directly optimizing concept coupling, resulting in greater performance.

We also conduct a study on human preferences regarding the generated results, as shown in Tab. 1. Specifically, annotators are presented with quadruplets consisting of (prompt, reference images, method 1 result, method 2 result) and are asked to select the better generation result based on two key criteria: (i) fidelity to the personalized subject or style, and (ii) alignment with the text prompt. The correspondence of method 1 (or 2) to either the compared method or our method is random-

---

[2]The "T" denotes text and the "I" denotes image, respectively.

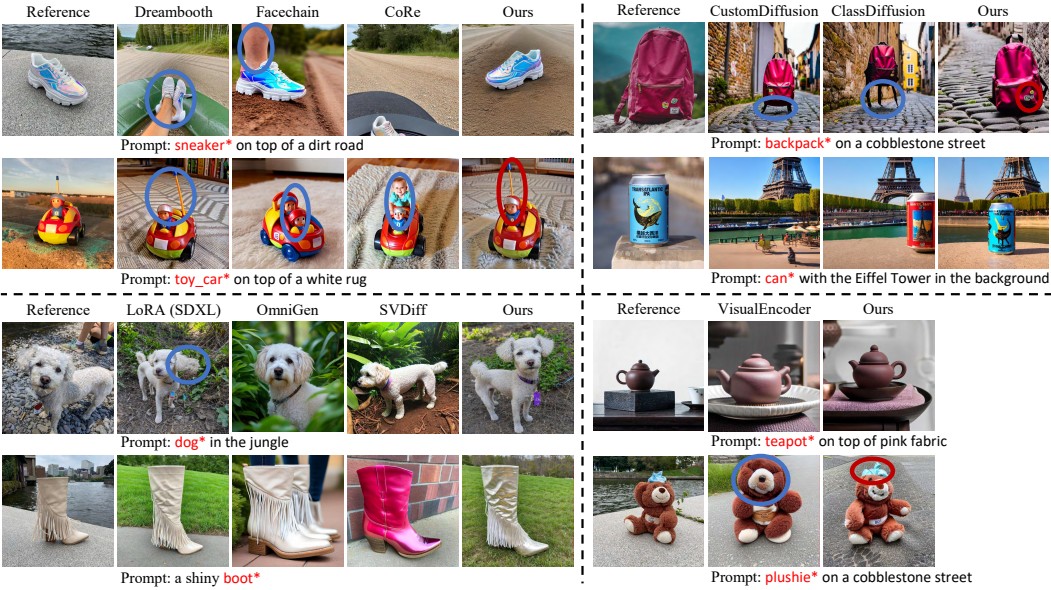

Figure 4: Subject personalization comparison across baselines, where **superclass\*** is the personalization target. One of multiple training references is shown. Red/blue circles highlight well-/poorly-generated regions. Our method achieves superior text alignment and personalization fidelity.

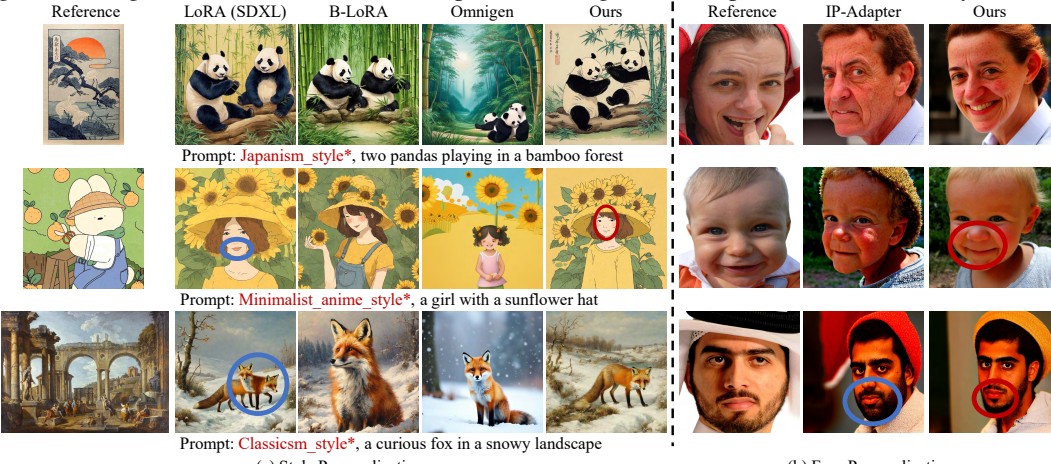

Figure 5: Comparison of style and face personalization results; **style\*** denotes the target style. For style personalization, the training set includes multiple references, and one is shown for brevity. Red circles highlight well-generated regions; blue circles mark areas with poor results. (a) Our model outputs styles closer to reference images: the Japanism result resembles a painting, the minimalist anime style result depicts the mouth as a line, and classicism result matches the original style without anomalies. (b) IP-Adapter alters gender (row 1) or makes faces appear older (row 2). Our method better replicates details such as beards (row 3).

ized and anonymized. We collect feedback from multiple annotators, resulting in a total of 1,800 responses. From this study, we observe that: (i) **Our method is generally preferred by users compared to all baselines**; and (ii) Notably, the greater the improvement in objective metrics over the baseline provided by our method, the more it is preferred by users, indicating an **alignment between subjective and objective evaluations**.

**Style Personalization.** We additionally compare with B-LoRA, which is specifically designed for style transfer by consolidating the training of two blocks for separating style and content. Tab.2 and Fig.5(a) show that our DDLoss and PDLoss significantly improve style personalization and boost all methods in a plug-and-play fashion. Similarly, LoRA (SDXL) with ACCORD, with 93M trainable parameters, outperforms Omnigen with 3.8B.

Table 3: Ablation study on the effects of DDLoss, and PDLoss across backbones.

| Method | CLIP-T | BLIP-T | CLIP-I | DINO-I |
|---|---|---|---|---|
| VE (SD1.5) | 25.9 | 36.1 | 79.1 | 75.5 |
| +PDLoss | 26.2 | 35.9 | 80.0 | 75.9 |
| +DDLoss | 26.0 | 35.8 | 79.8 | 75.8 |
| +PD & DDLoss | **26.3** | **36.1** | **80.4** | **76.7** |
| VE (SDXL) | 27.1 | 38.4 | 82.8 | 77.6 |
| +PDLoss | 27.8 | 39.5 | 82.9 | 77.4 |
| +DDLoss | 28.0 | **40.0** | 82.6 | 77.9 |
| +PD & DDLoss | **28.3** | 39.8 | **83.1** | **78.1** |
| LoRA (SD1.5) | 31.1 | 42.6 | 78.4 | 74.6 |
| +DDLoss | **31.8** | **43.0** | **78.4** | **75.1** |
| **LoRA (FLUX)** | 33.4 | 46.8 | 75.8 | 72.8 |
| +DDLoss | **34.8** | **47.8** | **78.2** | **73.4** |

Table 4: Impact of Reference Image Count on Subject-driven Personalization Performance.

| Method (Image Count) | CLIP-T | BLIP-T | CLIP-I | DINO-I |
|---|---|---|---|---|
| VE (1) | **25.0** | **34.2** | 75.9 | 71.0 |
| VE + Ours (1) | 24.7 | 33.3 | **78.9** | **73.9** |
| VE (3) | 25.0 | 34.5 | 78.0 | 74.3 |
| VE + Ours (3) | **25.6** | **34.8** | **79.4** | **75.7** |
| VE (all) | 25.9 | 36.1 | 79.1 | 75.5 |
| VE + Ours (all) | **26.3** | **36.1** | **80.4** | **76.7** |

Table 5: Quantitative results on FFHQ.

| Method | CLIP-T↑ | BLIP-T↑ | Face-Sim↑ |
|---|---|---|---|
| IP-Adapter | 20.0 | 34.7 | 14.8 |
| + Ours | **20.7** (+0.7) | **34.8** (+0.1) | **16.4** (+1.6) |

**Face Personalization.** We validate the potential of concept decoupling for zero-shot personalization, with a specific focus on face personalization, using the FFHQ dataset. Following the well-known zero-shot face personalization method IP-Adapter, we train the model with and without AC-CORD based on SD 1.5. Experimental results are shown in Tab. 5 and Fig. 5(b), demonstrating that the introduction of DDLoss and PDLoss simultaneously enhances face similarity and text alignment.

## 4.2 ABLATION STUDY

We study the impact of the proposed PDLoss and DDLoss on DreamBench in Tab. 3, and also investigate the impact of the number of reference images in Tab. 4. Indeed, the proposed loss functions work synergistically and hold regardless of the number of reference images and T2I backbone (including **FLUX**). Crucially, these studies confirm that both DDLoss and PDLoss contribute positively to performance (Tab. 3) and that our method remains effective even with a single reference image (Tab. 4), underscoring the robustness of our approach.

Table 6: Ablation Study on DDLoss and PDLoss Weights.

| Loss Weights | CLIP-T | BLIP-T | CLIP-I | DINO-I |
|---|---|---|---|---|
| CustomDiffusion (CD) | 34.2 | 45.4 | 62.7 | 56.9 |
| +0.1DD + 0.001PD | 33.9 | 46.5 | 70.7 | 64.9 |
| +0.1DD + 0.002PD | 34.0 | 46.5 | 70.5 | 64.8 |
| +0.1DD + 0.003PD | 33.9 | 46.4 | 71.1 | 65.2 |
| +0.2DD + 0.001PD | 33.9 | 46.5 | 70.7 | 64.8 |
| +0.2DD + 0.002PD | 33.9 | 46.5 | 70.8 | 65.1 |
| +0.2DD + 0.003PD | 34.0 | 46.6 | 70.7 | 65.0 |
| +0.3DD + 0.001PD | 33.9 | 46.4 | 71.0 | 65.3 |
| +0.3DD + 0.002PD | 34.0 | 46.5 | 70.8 | 65.1 |
| +0.3DD + 0.003PD | 34.0 | 46.5 | 70.7 | 64.9 |

We further study the effect of combining DDLoss and PDLoss with different weights on CustomDiffusion (CD) in Tab. 6. Introducing DDLoss with weights between 0.1 and 0.3, and PDLoss with weights between 0.001 and 0.003, consistently yields robust improvements across all metrics. This indicates that the performance of DDLoss and PDLoss is not sensitive to the precise choice of weights within these ranges. While variations in the weights have minimal impact on text alignment, we find that assigning a relatively larger weight to one loss and a smaller value to the other (e.g., 0.1DD + 0.003PD or 0.3DD + 0.001PD) is generally more beneficial for personalization fidelity than either both being too small or both being too large. This may be because both losses being small provide insufficient constraint on dependence discrepancy, while both being large excessively emphasize the auxiliary objectives and might hurt the diffusion goal.

## 5 CONCLUSION

This paper tackles concept coupling in image personalization by reframing it as a statistical dependency problem. We identify two distinct sources—a Denoising Dependence Discrepancy and a Prior Dependence Discrepancy—and introduce two corresponding plug-and-play losses, DDLoss and PDLoss, to directly mitigate them. Comprehensive experiments demonstrate that our method, ACCORD, successfully improves the critical balance between personalization fidelity and text control, offering a readily-integrable solution for a wide range of existing methods.

# 6 ACKNOWLEDGEMENT

This work was supported by Ant Group Research Intern Program.

# 7 REPRODUCIBILITY STATEMENT

To improve reproducibility, for theoretical results, such as Theorem 2, we provide the proofs in Appendix B. On the other hand, for experimental results, we provide the implementation details in Sec. 4, Appendix M and N. The VLM prompt used for generating image captions is also specified in Appendix O.

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

## A  APPENDIX

## APPENDIX

### A  PROOF OF THEOREM 1

We begin by briefly reviewing Theorem 1. The left-hand side (LHS) of Eq. (4) can be decomposed into the two terms as in Eq. (5):

$$\mathbb{E}_{\mathbf{x}_\theta}[|\log r(\mathbf{c}_p, \mathbf{c}_g | \mathbf{x}_{\theta,0}) - \log r(\mathbf{c}_s, \mathbf{c}_g)|]$$

Table 7: Meanings of notations.

| Notation | Meaning |
| --- | --- |
| $t$ | Denoising time step, ranging from $0$ to $T$. |
| $\mathbf{x}_0$ | Clear image or its latent code. |
| $\mathbf{x}_t$ | Noisy image or its latent code at time step $t$. |
| $\mathbf{x}_T$ | Noisy image or its latent code at time step $T$, modeled as a multivariate standard Gaussian noise. |
| $\alpha_t$ | Retention ratio of the original image at forward time step $t$. |
| $\boldsymbol{\epsilon}$ | Multivariate standard Gaussian noise. |
| $\theta$ | Network parameters. |
| $\boldsymbol{\sigma}_t$ | Standard deviation of the noisy code at time step $t$. |
| $\mathcal{U}_\theta(\mathbf{x}_t, \mathbf{c}, t)$ | Output of the denoising model at time step $t-1$ given generation condition $\mathbf{c}$. |
| $\mathbf{x}_{\theta,t}$ | Shorthand for denoising output at time step $t-1$ given generation condition $(\mathbf{c}_p, \mathbf{c}_g)$. |
| $\mathbb{D}$ | Training set for the image personalization task. |
| $\mathbf{x}^i$ | $i$-th reference image in the training set. |
| $\mathbf{c}^i$ | $i$-th generation condition in the training set. |
| $\mathbf{c}_p$ | Personalized target condition. |
| $\mathbf{c}_g$ | General text condition. |
| $\mathbf{c}_s$ | Text condition for the superclass of $\mathbf{c}_p$. |
| $r(\mathbf{c}_p, \mathbf{c}_g|\mathbf{x}_{\theta,t})$ | Conditional dependence coefficient for concepts $\mathbf{c}_p$ and $\mathbf{c}_g$ given generated image $\mathbf{x}_{\theta,t}$. |
| $r(\mathbf{c}_p, \mathbf{c}_g)$ | Prior dependence coefficient for concepts $\mathbf{c}_p$ and $\mathbf{c}_g$. |
| $\mathbf{f}_p, \mathbf{f}_s, \mathbf{f}_g$ | Projections using the CLIP Projector for $\mathbf{c}_p$, $\mathbf{c}_s$, and $\mathbf{c}_g$. |

$$= \mathbb{E}_{\mathbf{x}_\theta}\left[|\underbrace{\log r(\mathbf{c}_p, \mathbf{c}_g|\mathbf{x}_{\theta,0}) - \log r(\mathbf{c}_p, \mathbf{c}_g|\mathbf{x}_T)}_{\text{① Denoising Dependence Discrepancy}} + \underbrace{\log r(\mathbf{c}_p, \mathbf{c}_g) - \log r(\mathbf{c}_s, \mathbf{c}_g)}_{\text{② Prior Dependence Discrepancy}}|\right], \quad (13)$$

where $\mathbf{x}_T$ denotes multivariate standard Gaussian noise.

Since $\mathbf{x}_T$ is sampled independently of the conditions $\mathbf{c}_p$ and $\mathbf{c}_g$, it follows that $p(\mathbf{c}|\mathbf{x}_T) = p(\mathbf{c})$. Consequently,

$$\log \frac{p(\mathbf{c}_p, \mathbf{c}_g|\mathbf{x}_T)}{p(\mathbf{c}_p|\mathbf{x}_T)p(\mathbf{c}_g|\mathbf{x}_T)} = \log \frac{p(\mathbf{c}_p, \mathbf{c}_g)}{p(\mathbf{c}_p)p(\mathbf{c}_g)}. \quad (14)$$

Thus, the proof is complete.

## B  PROOF OF THEOREM 2

According to the definition of $r(\mathbf{c}_p, \mathbf{c}_g|\mathbf{x}_{\theta,t-1})$:

$$r(\mathbf{c}_p, \mathbf{c}_g|\mathbf{x}_{\theta,t-1}) = \frac{p(\mathbf{c}_p, \mathbf{c}_g|\mathbf{x}_{\theta,t-1})}{p(\mathbf{c}_p|\mathbf{x}_{\theta,t-1})p(\mathbf{c}_g|\mathbf{x}_{\theta,t-1})}, \quad (15)$$

the core of computing $\log r(\mathbf{c}_p, \mathbf{c}_g|\mathbf{x}_{\theta,t-1})$ lies in the computation of $p(\hat{\mathbf{c}}|\mathbf{x}_{\theta,t-1})$, where $\hat{\mathbf{c}}$ is an arbitrary condition. By applying Bayes' theorem, we have:

$$p(\hat{\mathbf{c}}|\mathbf{x}_{\theta,t-1}) = p(\hat{\mathbf{c}}|\mathbf{x}_{\theta,t-1}, \mathbf{x}_{\theta,t}) = \frac{p(\hat{\mathbf{c}}|\mathbf{x}_{\theta,t})p(\mathbf{x}_{\theta,t-1}|\mathbf{x}_{\theta,t}, \hat{\mathbf{c}})}{p(\mathbf{x}_{\theta,t-1}|\mathbf{x}_{\theta,t})}. \quad (16)$$

The first equation holds because the computation of $\mathbf{x}_{\theta,t-1}$ relies on $\mathbf{x}_{\theta,t}$:

$$\mathbf{x}_{\theta,t-1} = \mathcal{U}_\theta(\mathbf{x}_t, (\mathbf{c}_p, \mathbf{c}_g), t), \quad \mathbf{x}_t = \mathbf{x}_{\theta,t} + \boldsymbol{\sigma}_{t+1}\boldsymbol{\epsilon}_{t+1}, \quad \boldsymbol{\epsilon}_{t+1} \sim \mathcal{N}(0, I), \quad (17)$$

where $\boldsymbol{\sigma}_{t+1}$ is the standard deviation of the noisy code at time step $t+1$.

Next, we compute $p(\mathbf{x}_{\theta,t-1}|\mathbf{x}_{\theta,t}, \hat{\mathbf{c}})$ and $p(\mathbf{x}_{\theta,t-1}|\mathbf{x}_{\theta,t})$. In diffusion models, $p(\mathbf{x}_{\theta,t-1}|\mathbf{x}_{\theta,t}, \hat{\mathbf{c}})$ is a Gaussian distribution that can be parameterized as:

$$p(\mathbf{x}_{\theta,t-1}|\mathbf{x}_{\theta,t}, \hat{\mathbf{c}}) = \mathcal{N}\left(\mathbf{x}_{\theta,t-1}; \mathcal{U}_\theta(\mathbf{x}_{\theta,t}, \hat{\mathbf{c}}, t), \sigma_t^2\boldsymbol{I}\right) = \exp(C - \frac{\|\mathbf{x}_{\theta,t-1} - \mathcal{U}_\theta(\mathbf{x}_{\theta,t}, \hat{\mathbf{c}}, t)\|^2}{2\boldsymbol{\sigma}_t^2}), \quad (18)$$

where $C$ is a constant. We then substitute Eq. (17) into Eq. (18) and obtain:

$$p(\mathbf{x}_{\theta,t-1}|\mathbf{x}_{\theta,t}, \hat{\mathbf{c}}) = \exp(C - \frac{\|\mathcal{U}_\theta(\mathbf{x}_t, (\mathbf{c}_p, \mathbf{c}_g), t) - \mathcal{U}_\theta(\mathbf{x}_{\theta,t}, \hat{\mathbf{c}}, t)\|^2}{2\boldsymbol{\sigma}_t^2}), \quad (19)$$

Note that $\hat{\mathbf{c}}$ is an arbitrary condition, so $p(\mathbf{x}_{\theta,t-1}|\mathbf{x}_{\theta,t})$ can be obtained by setting $\hat{\mathbf{c}} = \varnothing$. Therefore, we substitute Eq. (19) into Eq. (16) and obtain:

$$\log p(\hat{\mathbf{c}}|\mathbf{x}_{\theta,t-1}) - \log p(\hat{\mathbf{c}}|\mathbf{x}_{\theta,t})$$

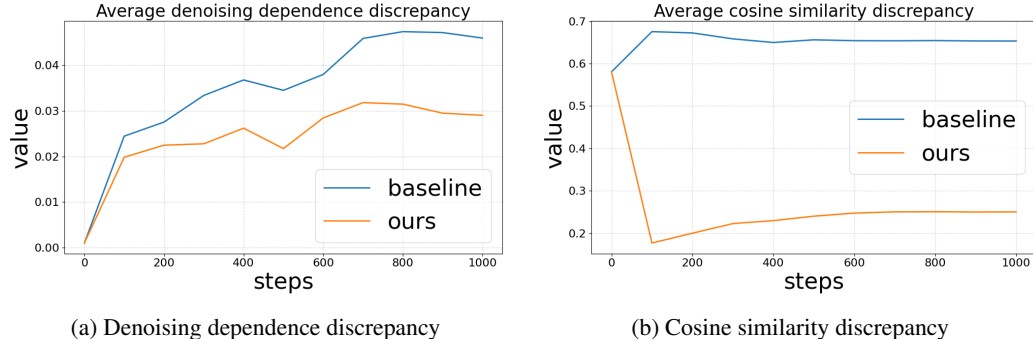

(a) Denoising dependence discrepancy

(b) Cosine similarity discrepancy

Figure 6: Visualization of the impact of DDLoss and PDLoss.

$$= \frac{1}{2\sigma_t^2}\Big[\|\mathcal{U}_\theta(\mathbf{x}_t,(\mathbf{c}_p,\mathbf{c}_g),t) - \mathcal{U}_\theta(\mathbf{x}_{\theta,t},\varnothing,t)\|^2 - \|\mathcal{U}_\theta(\mathbf{x}_t,(\mathbf{c}_p,\mathbf{c}_g),t) - \mathcal{U}_\theta(\mathbf{x}_{\theta,t},\hat{\mathbf{c}},t)\|^2\Big] \quad (20)$$

Finally, by substituting Eq. (20) into the definition of $r(\mathbf{c}_p,\mathbf{c}_g|\mathbf{x}_{\theta,t-1})$ (15), we obtain:

$$\log r(\mathbf{c}_p,\mathbf{c}_g|\mathbf{x}_{\theta,t-1}) - \log r(\mathbf{c}_p,\mathbf{c}_g|\mathbf{x}_{\theta,t})$$
$$= \frac{1}{2\sigma_t^2}\Big[\|\mathcal{U}_\theta\big(\mathbf{x}_t,(\mathbf{c}_p,\mathbf{c}_g),t\big) - \mathcal{U}_\theta\big(\mathbf{x}_{\theta,t},\mathbf{c}_p,t\big)\|^2$$
$$+ \|\mathcal{U}_\theta\big(\mathbf{x}_t,(\mathbf{c}_p,\mathbf{c}_g),t\big) - \mathcal{U}_\theta\big(\mathbf{x}_{\theta,t},\mathbf{c}_g,t\big)\|^2$$
$$- \|\mathcal{U}_\theta\big(\mathbf{x}_t,(\mathbf{c}_p,\mathbf{c}_g),t\big) - \mathcal{U}_\theta\big(\mathbf{x}_{\theta,t},\varnothing,t\big)\|^2\Big]. \quad (21)$$

This completes the proof.

Table 8: Ablation study on the PDLoss design.

| Optimization target | CLIP-T↑ | BLIP-T↑ | CLIP-I↑ | DINO-I↑ |
|---|---|---|---|---|
| VisualEncoder wo/ Ours | 25.9 | 36.1 | 79.1 | 75.5 |
| $\mathbb{E}_{\mathbf{c}_g}\big[|cos(\mathbf{f}_p,\mathbf{f}_g) - cos(\mathbf{f}_s,\mathbf{f}_g)|\big]$ | 26.2 (+0.3) | 35.9 (-0.2) | **80.0** (+0.9) | **75.9** (+0.4) |
| $\mathbb{E}_{\mathbf{c}_g}\big[|cos(\mathbf{f}_p,\mathbf{f}_g)|\big]$ | 26.4 (+0.4) | 36.8 (+0.7) | 79.9 (+0.8) | 75.5 (+0.0) |
| $\mathbb{E}_{\mathbf{c}_g}\big[|cos(\mathbf{f}_p,\mathbf{f}_g) + 1|\big]$ | **27.7** (+1.8) | **38.4** (+2.3) | 77.6 (-1.5) | 73.3 (-2.2) |
| $\mathbb{E}_{\mathbf{c}_g}\big[|1 - \cos(\mathbf{f}_p - \mathbf{f}_g, \mathbf{f}_s - \mathbf{f}_g)|\big]$ | 26.5 | 36.9 | 79.5 | 75.5 |

## C   IMPACT OF DDLOSS AND PDLOSS IN REDUCING DEPENDENCE DISCREPANCY

To clearly demonstrate the roles of DDLoss and PDLoss during training, we visualize their effects in Fig. 6. It can be observed that with the use of DDLoss, the increase in denoising dependence discrepancy, $|\log r(\mathbf{c}_p,\mathbf{c}_g|\mathbf{x}_{\theta,0}) - \log r(\mathbf{c}_p,\mathbf{c}_g|\mathbf{x}_T)|$, is suppressed. On the other hand, the application of PDLoss results in a reduction in the cosine similarity discrepancy $|cos(\mathbf{f}_p,\mathbf{f}_g) - cos(\mathbf{f}_s,\mathbf{f}_g)|$.

## D   ABLATION STUDY ON THE IMPACT OF PDLOSS DESIGN

To minimize concept coupling in Eq. (4):

$$\mathbb{E}_{\mathbf{x}_\theta}[|\log r(\mathbf{c}_p,\mathbf{c}_g|\mathbf{x}_{\theta,0}) - \log r(\mathbf{c}_s,\mathbf{c}_g)|], \quad (22)$$

we align the cosine similarity $cos(\mathbf{f}_p,\mathbf{f}_g)$ with $cos(\mathbf{f}_s,\mathbf{f}_g)$ in Eq. (11).

$$\mathcal{L}_{\text{PD}} = \mathbb{E}_{\mathbf{c}_g}[|cos(\mathbf{f}_p,\mathbf{f}_g) - cos(\mathbf{f}_s,\mathbf{f}_g)|], \quad (23)$$

To further understand the role of the cosine similarity target in PDLoss, we study its impact in Tab. 8. In addition, we also compare our PDLoss with an empirical design: $\mathbb{E}_{\mathbf{c}_g}[|1 - \cos(\mathbf{f}_p - \mathbf{f}_g, \mathbf{f}_s - \mathbf{f}_g)|]$. It is observed that: (i) As the cosine similarity target decreases, metrics related to text alignment, namely CLIP-T and BLIP-T, improve, whereas metrics associated with personalization fidelity, such as CLIP-I and DINO-I, decline. This observation aligns with our derivation.

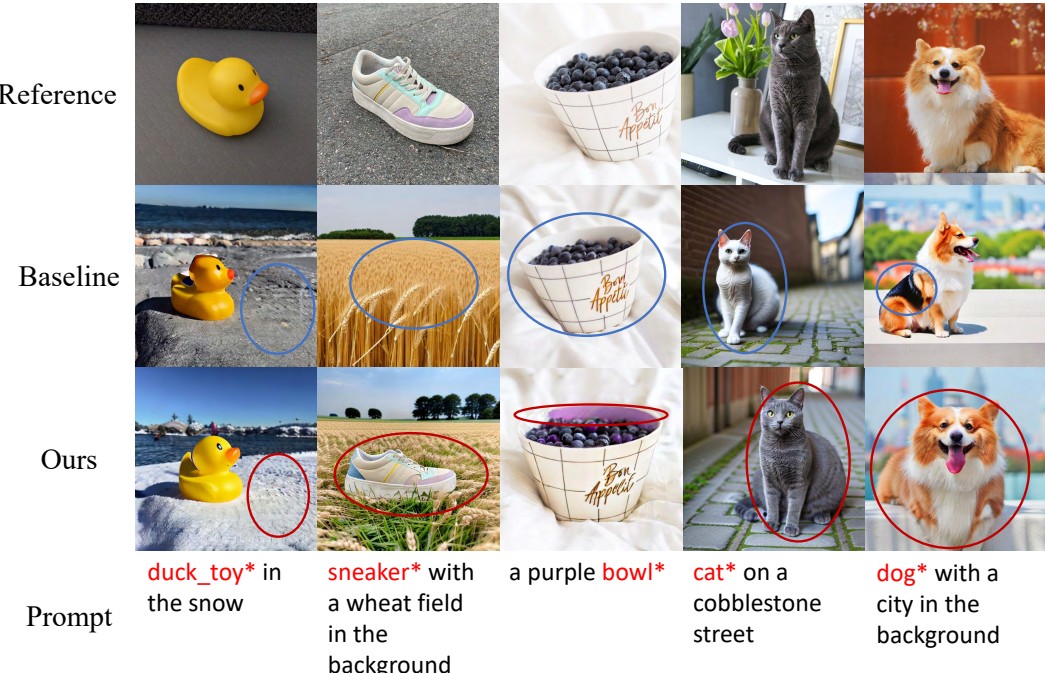

Figure 7: A comparison of the visual outcomes of subject personalization, where "**superclass***" denotes the personalization target. In the 1st, 2nd, and 3rd columns, our method aligns better with the prompt and successfully generates a snowy scene, a wheat field, and a bowl with an inner purple wall; in contrast, the baseline model fails to do so. In the 4th and 5th columns, our method generates subjects that bear a closer resemblance to the reference images. However, the baseline either produces an unrelated cat (4th column) or generates anomalies like a black dog's back (5th column). It should be noted that for columns 1, 2, and 5, our method not only replaces the background but also adjusts the perspective to make the generated image look more natural.

A lower cosine similarity indicates a reduced $p(\mathbf{c}_g|\mathbf{c}_p)$, implying that $\mathbf{c}_p$ is less likely to interfere with other text concepts. However, if the similarity between $\mathbf{c}_p$ and $\mathbf{c}_g$ decreases excessively, it becomes challenging for $\mathbf{c}_p$ to maintain inherent relationships with its superclass and other concepts, thereby impairing personalization fidelity. Consequently, setting the cosine similarity target as $cos(\mathbf{f}_s, \mathbf{f}_g)$ achieves a balance between text alignment and personalization fidelity. (ii) The empirical approach, $\mathbb{E}_{\mathbf{c}_g}[\|1 - \cos(\mathbf{f}_p - \mathbf{f}_g, \mathbf{f}_s - \mathbf{f}_g)\|]$, also improves upon the baseline by emphasizing text alignment. However, this method cannot be derived from Eq. (9), namely the definition of prior dependence discrepancy.

$$\log r(\mathbf{c}_p, \mathbf{c}_g) - \log r(\mathbf{c}_s, \mathbf{c}_g) = \log \frac{p(\mathbf{c}_g|\mathbf{c}_p)}{p(\mathbf{c}_g|\mathbf{c}_s)}. \tag{24}$$

## E   APPLICATION ON MULTI-SUBJECT PERSONLIZATION

Table 9: Performance of ACCORD on multi-subject personalization by integrating into Break-A-Scene (Avrahami et al. (2023)).

| Method | CLIP-T | BLIP-T | CLIP-I | DINO-I |
|---|---|---|---|---|
| Break-A-Scene | 31.1 | 42.0 | 51.6 | 36.1 |
| w/ Ours | **31.2** | **42.0** | **53.2** | **38.7** |

We explore the compatibility of ACCORD with multi-subject personalization methods. Break-A-Scene is a well-known multi-subject personalization method that achieves disentanglement of multiple subjects by randomly sampling subject combinations, computing diffusion loss based on explicit masks, and constraining the cross-attention map. Notably, the DDLoss and PDLoss proposed by

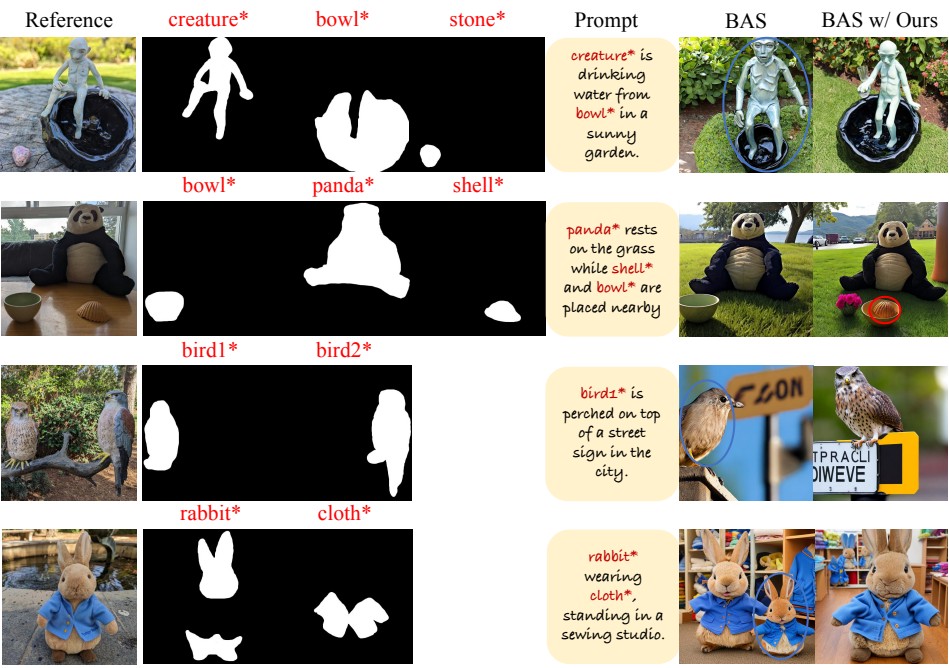

Figure 8: Qualitative comparison of multi-subject generation results, where 'BAS' is short for Break-A-Scene. ACCORD enhances the personalization fidelity of Break-A-Scene in the first and third rows. In the second row, ACCORD correctly generates a shell as specified by the prompt, and in the fourth row, unlike Break-A-Scene, it does not generate two rabbits.

ACCORD are compatible with Break-A-Scene. Therefore, we incorporate ACCORD into Break-A-Scene and conduct comparative experiments on its example dataset. The prompts used for evaluation are constructed by GPT, and include both single-subject generation (i.e., generating one of the multiple subjects present in the training image) and compositional multi-subject generation. As shown in Tab. 9, ACCORD further improves concept decoupling capabilities on top of Break-A-Scene. The personalization fidelity metrics CLIP-I and DINO-I are significantly improved, suggesting that concept decoupling facilitates the model's ability to better capture the appearances of distinct subjects. As illustrated in Fig. 8, ACCORD enhances the personalization fidelity of Break-A-Scene in the first and third rows. In the second row, ACCORD correctly generates a shell as specified by the prompt, and in the fourth row, unlike Break-A-Scene, it does not generate two rabbits.

## F SENSITIVITY TO VLM-GENERATED CAPTIONS

Table 10: Sensitivity of ACCORD to VLM-generated Captions.

| Caption Source | CLIP-T | BLIP-T | CLIP-I | DINO-I |
|---|---|---|---|---|
| InternVL2-8B | 34.1 | 46.6 | 71.4 | 65.6 |
| Qwen3-VL-8B | 34.0 | 45.8 | 71.8 | 66.1 |
| GLM-4.5V-Thinking-9B | 33.9 | 46.1 | **72.9** | **67.9** |
| Human | **34.5** | **46.8** | 71.8 | 65.7 |

We assess the sensitivity of ACCORD to the quality of VLM-generated captions for subject personalization using CustomDiffusion, as shown in Tab. 10. While captions generated by different VLMs can influence the results to some extent, the degree of variation remains relatively small. Notably, captions generated by GLM-4.5V-Thinking-9B exhibit higher fidelity, even surpassing human-annotated captions. This suggests that employing automated VLM-based captions for ACCORD is both reasonable and effective.

We further analyze the generalizability of decoupling achieved by optimizing DDLoss and PDLoss. We observe from our existing results that, in some cases, even when a concept was not specified in

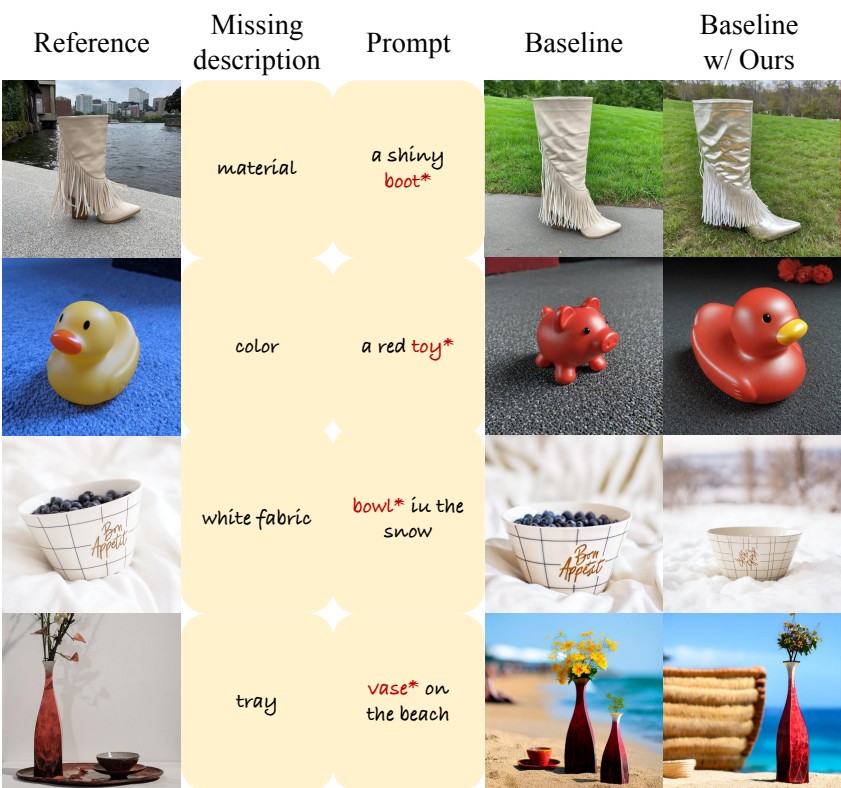

Figure 9: Generalization of concept decoupling in ACCORD via DDLoss and PDLoss optimization. 'Missing description' indicates that relevant information is absent or removed from the training prompts, whereas 'Prompt' refers to those used for image generation. Even in cases where descriptions of material, color, or other objects are absent from the training prompts, ACCORD may also be able to decouple these concepts by generalizing the relationships between the personalization target's superclass and other concepts.

the prompt, ACCORD may still able to decouple it. To further test this, we conduct an experiment in which we remove a particular concept from the prompts within the training set, despite the concept being present in the corresponding images. We then train ACCORD on this modified dataset and evaluate whether the generated images suppress the coupling between the personalization target and the excluded concepts. The results are shown in Fig. 9. It is observed that even in cases where descriptions of material, color, or other objects are absent from the training prompts, ACCORD may also be able to decouple these concepts by generalizing the relationships between the personalization target's superclass and other concepts.

## G    PROMPT-BASED SELECTIVE PERSONALIZATION

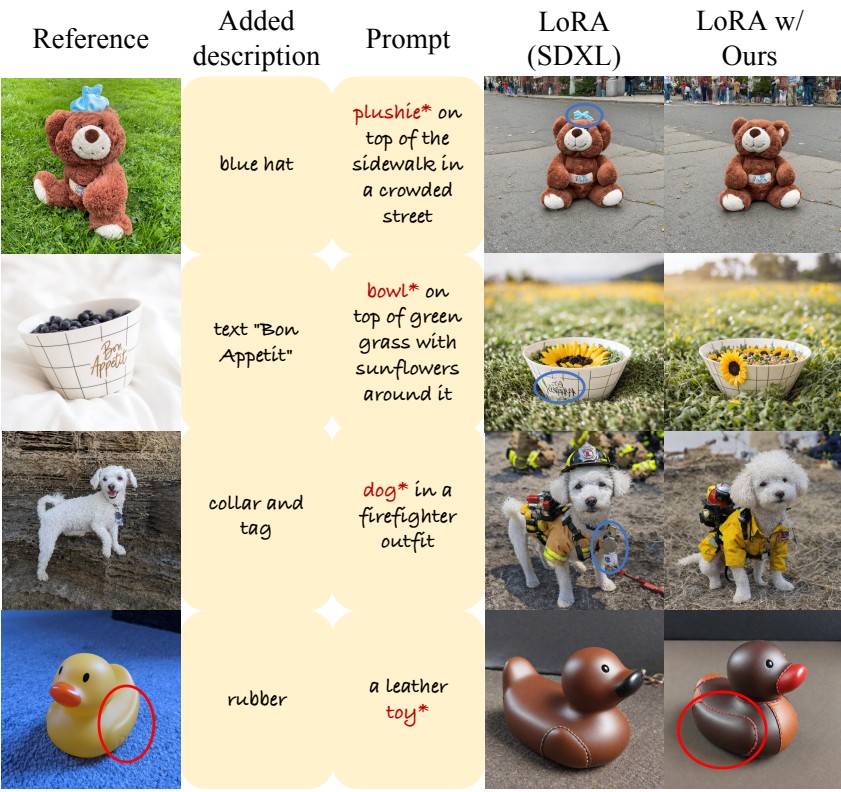

Figure 10: Qualitative exploration on selective attribute decoupling in subject personalization using ACCORD. 'Added description' denotes the inclusion of elements not intended for personalization in the training prompts, whereas 'Prompt' refers to those used for image generation. ACCORD successfully prevents the personalization of the blue hat on the teddy bear's head, the lettering on the bowl's side, and the collar and tag on the dog's neck, and it yields better result of the leather duck toy.

The ACCORD's decoupling between the personalization target and other conditions in the prompt provides users with the flexibility to specify concepts they do not wish to personalize, especially when such concepts are difficult to remove from the images. We qualitatively explore this capability by selecting subjects with multiple attributes from DreamBench and manually including certain attributes that were not intended for personalization in the corresponding prompts. Subject personalization experiments are then conducted on this dataset, and the results are presented in Fig. 10. As shown in the figure, when attributes that are not intended to be personalized are included in the prompts, ACCORD successfully avoids personalizing these attributes. This may be useful in scenarios where it is desirable to modify certain properties of the subject without damaging the subject to be personalized.

Table 11: Performance of ACCORD by prompt category on DreamBench.

| Method | Prompt Type | Percentage | CLIP-T | BLIP-T | CLIP-I | DINO-I |
|---|---|---|---|---|---|---|
| CustomDiffusion | Background Replacement | 24% | 35.1 | 47.0 | 65.2 | 58.8 |
| | Situational Placement | 32% | 35.7 | 47.2 | 65.8 | 59.9 |
| | Object Composition | 24% | 33.9 | 45.8 | 62.9 | 55.9 |
| | Attribute Editing | 20% | 31.0 | 40.0 | 54.3 | 51.0 |
| ACCORD | Background Replacement | 24% | 35.4 | 48.5 | 75.2 | 69.0 |
| | Scene Placement | 32% | 35.1 | 48.2 | 73.9 | 67.6 |
| | Object Composition | 24% | 33.7 | 46.9 | 71.3 | 64.4 |
| | Attribute Editing | 20% | 31.3 | 41.2 | 63.0 | 59.8 |

## H  PERFORMANCE BY PROMPT CATEGORY ON DREAMBENCH

To better understand the source of performance improvement by concept decoupling, we categorize the validation prompts in DreamBench and report the performance of ACCORD and the baseline, CustomDiffusion, for each category, as shown in Tab. 11. Specifically, the prompts in DreamBench can be grouped into four categories: (i) **Background Replacement**, e.g., "$c_p$ with a city in the background," where $c_p$ denotes the personalization target; (ii) **Situational Placement**, e.g., "$c_p$ on top of green grass with sunflowers around it"; (iii) **Object Composition**, e.g., "$c_p$ floating in an ocean of milk"; and (iv) **Attribute Editing**, e.g., "a shiny $c_p$." We find that ACCORD delivers balanced performance improvements across different prompt types, with slightly higher gains in background replacement and attribute editing, which demand a higher degree of concept decoupling.

## I  SENSITIVITY TO TIMESTEP SAMPLING AND CLASSIFIER-FREE GUIDANCE SCALE

Table 12: Sensitivity of ACCORD to timestep sampling strategies.

| Method | Timestep Sampling | CLIP-T | BLIP-T | CLIP-I | DINO-I |
|---|---|---|---|---|---|
| CustomDiffusion | Uniform | 34.2 | 45.4 | 62.7 | 56.9 |
| | LogitNormal(0.0, 1.0) | 34.0 | 45.4 | 63.1 | 57.5 |
| | LogitNormal(1.0, 0.6) | 34.3 | 45.7 | 61.3 | 55.0 |
| ACCORD | Uniform | 34.1 | 46.6 | 71.4 | 65.6 |
| | LogitNormal(0.0, 1.0) | 34.1 | 46.4 | 71.6 | 66.4 |
| | LogitNormal(1.0, 0.6) | 34.2 | 46.5 | 69.3 | 63.1 |

Table 13: Sensitivity of ACCORD to Classifier-Free Guidance (CFG) scales.

| Method | CFG Scale | CLIP-T | BLIP-T | CLIP-I | DINO-I |
|---|---|---|---|---|---|
| CustomDiffusion | 5.0 | 33.7 | 45.1 | 61.5 | 55.8 |
| | 7.5 | 34.2 | 45.4 | 62.7 | 56.9 |
| | 10.0 | 34.0 | 45.2 | 61.2 | 54.3 |
| ACCORD | 5.0 | 33.8 | 46.3 | 70.7 | 65.5 |
| | 7.5 | 34.1 | 46.6 | 71.4 | 65.6 |
| | 10.0 | 34.1 | 46.3 | 70.2 | 64.6 |

We investigate the robustness of ACCORD to different timestep sampling strategies and guidance scales for subject personalization using CustomDiffusion, as shown in Tab. 12 and Tab. 13. The results demonstrate that ACCORD consistently improves upon the baseline and performs robustly across noise settings and guidance scales. This robustness can be attributed to the fact that neither DDLoss nor PDLoss makes assumptions regarding the noise setting or guidance scale. Specifically, DDLoss constrains the growth of dependency between neighboring timesteps without limiting timesteps themself, while PDLoss is independent of the denoising process.

## J  REAL WORLD PERSONALIZATION

We further collect a set of hamster photographs for personalization experiments, aiming to evaluate the practical effectiveness of our proposed method in real-world personalization scenarios. The

Reference      Prompts      LoRA (SDXL)      Ours

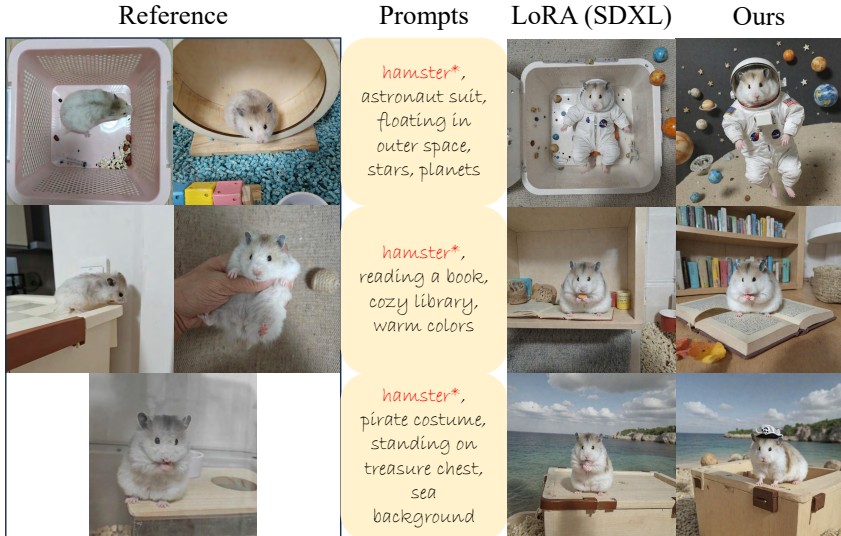

Figure 11: Real-world personalization visualization results. Compared with the baseline, our approach successfully generates stars (1st row), a library scene (2nd row), and a pirate hat (3rd row).

quantitative results and qualitative examples are presented in Tab. 14 and Fig. 11, respectively. It can be observed that our method substantially improves the text control capability over the baseline. As shown in Fig. 11, our approach successfully generates stars (1st row), a library scene (2nd row), and a pirate hat (3rd row).

Table 14: Real-world personalization quantitative results.

| Method | CLIP-T↑ | BLIP-T↑ | CLIP-I↑ | DINO-I↑ |
|---|---|---|---|---|
| LoRA (SDXL) | 38.6 | 52.1 | 68.9 | **59.1** |
| LoRA (SDXL) w/ Ours | **39.9** | **54.2** | **69.1** | 58.6 |

## K   MORE VISUALIZATION RESULTS AND FAILURE CASES

### K.1   ADDITIONAL VISUALIZATION RESULTS FOR SUBJECT, STYLE, AND FACE PERSONALIZATION.

We provide more visualization results in Fig. 7, 13 and 14. For subject and style personalization, the "Baseline" is Dreambooth. For face personalization, the "Baseline" is IP-Adapter. The following observations can be made: (1) Our method demonstrates superior text alignment compared to the baseline. Specifically, in the first, second, and third columns of Fig. 7, our method successfully generates a snowy scene, a wheat field, and a purple bowl, whereas the baseline model does not. In the first, third, fourth and fifth columns of Fig. 13, our approach successfully produces images of a pirate, a snowy landscape, a knight and a blue shield. Finally, in the third and fourth columns of Fig. 14, our method generates a cityscape background and cultural elements according to the prompts. (2) Our method better preserves personalization fidelity. In the fourth and fifth columns of Fig. 7, our method generates subjects that more closely resemble the reference images, whereas the baseline either produces an unrelated cat (4th column) or anomalies such as a black dog's back (5th column). It should be noted that for columns 1, 2, and 5, our method not only replaces the background but also adjusts the perspective to make the generated image look more natural. In the second row of Fig. 13, the images generated by our method exhibit styles more closely aligned with the reference styles, namely the clay style. Finally, in all columns of Fig. 14, the faces generated by our method more closely resemble the reference faces. Specifically, our method better captures the subject's age in columns 1 and 5; and the hair style in columns 2 and 3.

## K.2 VISUALIZATION OF THE EFFECTS OF PDLOSS AND DDLOSS.

We also visualize the individual effects of DDLoss and PDLoss on subject personalization based on CustomDiffusion in Fig. 12. It can be observed that PDLoss mainly improves personalization fidelity by aligning the relationship between the personalization target and other concepts with that of its superclass and other concepts. On the other hand, incorporating DDLoss enhances text control capabilities and can further boost personalization fidelity. Specifically, after introducing DDLoss, the bear in the first column of Fig.12 more closely resembles the reference image; meanwhile, the toys in the second and third columns are correctly placed on the sidewalk and in the jungle, respectively.

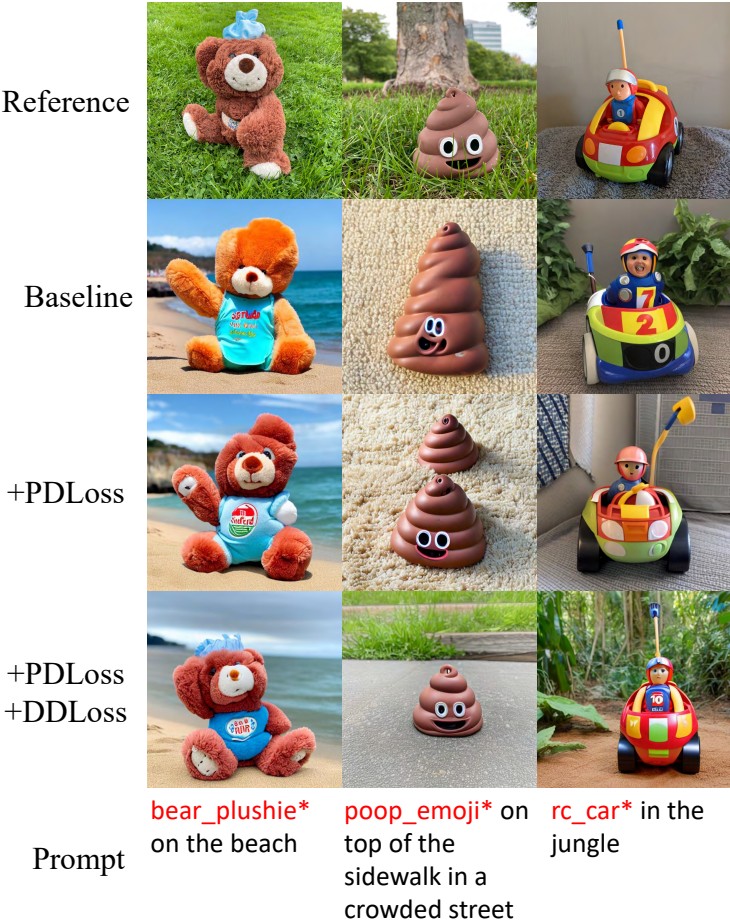

Figure 12: Visualization of the Individual Effects of DDLoss and PDLoss on Subject Personalization Based on CustomDiffusion. PDLoss aligns the relationships between the personalization target and other concepts with the relationships between its superclass and those concepts, thereby mainly enhancing fidelity. DDLoss further improves text alignment and can also boost fidelity. In the first column, adding DDLoss makes the generated bear more resemble the reference image. In the second and third columns, DDLoss enables the models to correctly place the personalized toys on the sidewalk and in the jungle, respectively.

## K.3 FAILURE CASE ANALYSIS.

Finally, we present several failure cases of ACCORD in Fig. 15. The causes of these failures can be broadly categorized into two types: (1) Concepts that are strongly entangled with the personalization target are not explicitly disentangled during training. If a concept that is undesirably coupled with the personalization target is not included in the training prompts, it will not be addressed by DDLoss and PDLoss. Consequently, at inference time, the model must rely solely on its generalization ability

to disentangle this concept from the personalization target, which may lead to failure. (2) Inaccurate modeling of concept dependencies by the foundation T2I model. The effectiveness of DDLoss and PDLoss is fundamentally constrained by the capabilities of the underlying T2I foundation model. When the base model struggles to simultaneously generate both the superclass of the personalization target and another concept, it likely fails to accurately capture the dependencies between them. In such cases, the decoupling effect of DDLoss becomes limited. This observation also suggests that more powerful foundation T2I models can enable DDLoss to achieve better disentanglement.

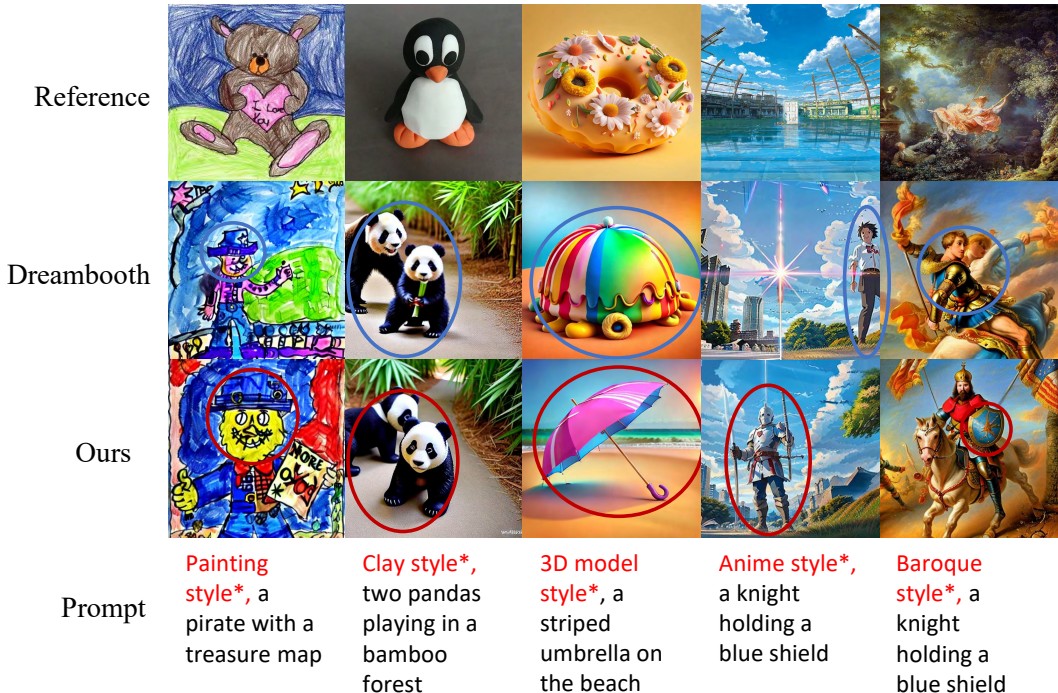

Figure 13: A comparison of style personalization visual outcomes, with "**style***" indicating the target style. Compared to the baseline, our method successfully generates a pirate, a snowy landscape, a knight, and a blue shield in columns 1, 3, 4, and 5. In column 2, our approach produces clay-style images that more closely match the reference.

## L  COMPUTATION AND MEMORY OVERHEAD

Our proposed ACCORD can be seamlessly incorporated into many existing image personalization methods, enhancing personalization performance at the expense of increased GPU memory usage and longer training times. Tab. 15 summarizes the GPU memory consumption and training duration for each baseline method and its integration with ACCORD, under consistent training settings: all experiments use a batch size of 4 and are trained for 1000 steps on an NVIDIA H100 GPU.

While integrating ACCORD introduces additional GPU memory requirements and slightly longer training times, these increases are moderate and not reach an order-of-magnitude larger compared to the respective baselines. Furthermore, we observe that reducing the batch size has a negligible impact on the performance of ACCORD, enabling users to lower batch size in practical scenarios to achieve acceptable memory usage and training time. Crucially, the extra computational cost imposed by ACCORD remains manageable, especially when contrasted with zero-shot approaches, which often involve considerably higher training overheads and less controllable personalization outcomes at inference time.

## M  DETAILED DATASET INFORMATION

We utilize the DreamBench Ruiz et al. (2023) dataset to compare the subject-driven personalization capabilities of different methods. DreamBench contains 30 subjects across 15 categories, of

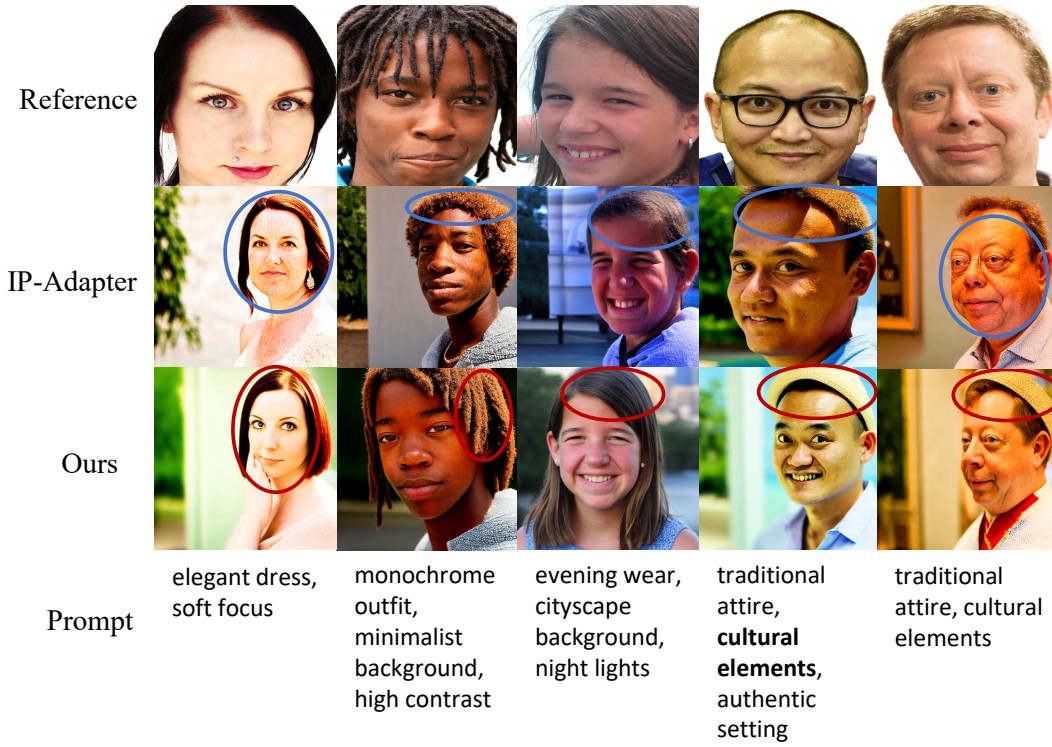

Figure 14: A comparison of the visual outcomes of face personalization. Red circles highlight well-generated areas, while blue circles indicate poorly generated regions. The baseline IP-Adapter tends to alter gender or make faces appear older in columns 1, 2, and 5. In contrast, our method produces faces more similar to the reference in columns 1, 2, 4, 5. Additionally, in columns 3 and 4 of Fig. 14, our model generates cityscape backgrounds and incorporates cultural elements according to the prompts.

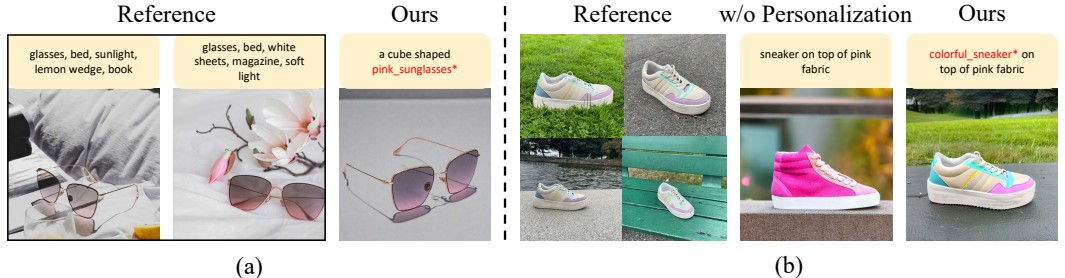

Figure 15: Failure cases of our ACCORD method. (a) Failure to generate cube-shaped sunglasses: When specific attributes are absent from the training prompts, they are omitted from the calculation of DDLoss and PDLoss. In such instances, disentanglement relies solely on the model's generalization ability, which can lead to incomplete or unsuccessful attribute decomposition. (b) Failure to generate the target sneaker on pink fabric: The effectiveness of DDLoss and PDLoss is inherently limited by the capacity of the base model. If the underlying T2I foundation model cannot generate both the personalization target's superclass and another concept simultaneously, it may fail to accurately model the dependence between these concepts. As a result, DDLoss's cross-timestep alignment mechanism may also fail to achieve proper disentanglement.

which 9 are animals, with each subject having 4-6 images. For style personalization, we employ StyleBench Junyao et al. (2024), which focuses on style transfer tasks and includes 73 distinct styles, each style comprising 5 or more reference images. Furthermore, to validate the effectiveness of our proposed losses for zero-shot image personalization, we conduct face personalization experiments

| Method | GPU Memory (GB) | Training Time (s) |
|---|---|---|
| DreamBooth | 26.5 | 320 |
| DB w/ Ours | 45.3 | 480 |
| CustomDiffusion | 18.7 | 346 |
| CD w/ Ours | 29.5 | 502 |
| LoRA (SDXL) | 27.7 | 483 |
| LoRA (SDXL) w/ Ours | 60.8 | 916 |
| VisualEncoder | 11.1 | 255 |
| VE w/ Ours | 16.8 | 490 |

Table 15: Computation and memory overhead for different methods and their integration with AC-CORD. All experiments are conducted with batch size 4 and 1000 training steps on H100.

on the FFHQ Karras et al. (2021) dataset. FFHQ is a dataset of 70,000 high-quality face images, offering substantial diversity in age, ethnicity, background, etc. We employ Insightface Deng et al. (2019) to detect over 40,000 images containing only a single face, and exclusively use these images for training and testing.

## N    MORE IMPLEMENTATION DETAILS

The baseline VisualEncoder Ye et al. (2023) is a simplified version of IP-Adapter that retains the CLIP Image Encoder-based Visual Encoder, omitting the image-specific Cross Attention. This design implies that only the MLP at the end of the CLIP Image Encoder is trainable, and the personalization relies entirely on the visual embeddings $c_p$ extracted by the visual encoder. We find that it serves as a strong parameter-efficient baseline. We utilize the official implementation of Facechain-SuDe while implementing other baselines and our proposed method using open-source library Diffusers von Platen et al. (2022). All methods employ the DDIM sampler, a guidance scale of 7.5, and 50 inference steps during evaluation. We conduct experiments on NVIDIA A100 and H100 GPUs.

The different training paradigms of the various baselines necessitate distinct weighting for DDLoss and PDLoss. After tuning the loss weights using validation prompts, we find that, in general, a DDLoss weight between 0.1 and 0.3 suffices, while a PDLoss weight between 0.001 and 0.003 is adequate. We train all methods for 1000 steps on each subject or style and display the results of the best-performing step. It is noteworthy that users can adjust the loss weights in practice to achieve optimal results due to the automatic computation of CLIP-T, BLIP-T, CLIP-I, DINO-I, Gram-D, and Face-Sim.

## O    VLM PROMPTS FOR IMAGE CAPTIONING

We employ Intern-VL2 Chen et al. (2024b) as the image captioner. The prompt used is detailed below:

```
You are an excellent prompt engineer. Given an image and a tag
    corresponding to an important object in the image, please describe
    the given image in short for the image generation process of the SD
    model.

Note that the prompt you give should consist of a series of phrases, not
    a complete sentence, and must contain the tag corresponding to the
    important object. Please do not describe the important object in
    detail. Please do not answer anything other than the prompt. The
    prompt you give needs to use all lowercase letters. Here is an
    example: 1 dog, running, sea, sunset.

Now, the important objects are:
```

## P  LIMITATIONS

Autoregressive generative models without a diffusion process, such as LlamaGen Sun et al. (2024), are not compatible with the proposed losses. Furthermore, the effectiveness of our decoupling losses is constrained by the capabilities of the foundation T2I model; if the base model cannot accurately represent the relationship between a superclass and a given concept, ACCORD's ability to regularize this dependency is limited. Finally, decoupling is most effective for concepts that are explicitly included in training prompts, while concepts that are implicitly coupled may not be fully disentangled.

## Q  LLM USAGE

In this paper, large language models (LLMs) are only used to assist or polish the writing, and they are not involved in the methodology and experimental design of this paper.

