# OpenReview forum: "ACCORD: Alleviating Concept Coupling through Dependence Regularization for Text-to-Image Diffusion Personalization"
_ICLR.cc/2026/Conference — ICLR 2026 Poster_

### Official Review · Reviewer_kN9B · 2025-10-21

**Soundness:** 3
**Presentation:** 3
**Contribution:** 3
**Rating:** 6
**Confidence:** 4

**Summary:**

The paper addresses concept coupling in T2I personalization and proposes ACCORD with two plug-and-play losses to control denoising/prior dependence. Results show consistent alignment/fidelity gains across several backbones with modest overhead.

**Strengths:**

1. Concept coupling is formalized as conditional dependence and decomposed into denoising vs. prior components, each tied to a dedicated loss.
2. Provides a closed/near-closed form via the “model-as-implicit-classifier” view—no auxiliary discriminator or teacher model required.

**Weaknesses:**

1. Some works need to be discussed and compared in the Related Work section，Update-space constraints: Custom Diffusion [1], PaRa [2], Attention-level preservation: Perfusion [3], Attend-and-Excite [4]. Embedding-only personalization: Textual Inversion [5].
2. Human-eval reporting lacks agreement stats



[1] Nupur Kumari, Bing Li, David Forsyth, Jia-Bin Huang, Vishal M. Patel, Oncel Tuzel, Ali Farhadi, Anima Anandkumar. Multi-Concept Customization of Text-to-Image Diffusion. CVPR, 2023.

[2] Shangyu Chen, Zizheng Pan, Jianfei Cai, Dinh Phung. PaRa: Personalizing Text-to-Image Diffusion via Parameter Rank Reduction. ICLR, 2025.

[3] Omer Tov, Yuval Alaluf, Yotam Nitzan, Daniel Cohen-Or, Tali Dekel. Key-Locked Rank-One Editing for Text-to-Image Personalization (Perfusion). SIGGRAPH, 2023.

[4] Hila Chefer, Yuval Alaluf, Yael Vinker, Lior Wolf, Daniel Cohen-Or. Attend-and-Excite: Attention-Based Semantic Guidance for Text-to-Image Diffusion Models. 2023.

[5] Rinon Gal, Yuval Alaluf, Yuval Atzmon, Or Patashnik, Amit H. Bermano, Gal Chechik, Daniel Cohen-Or. An Image is Worth One Word: Personalizing Text-to-Image Generation using Textual Inversion. 2022.

**Questions:**

1. How sensitive are your losses to scheduler/noise settings and guidance scales?
2. Can ACCORD be demonstrated in multi-concept personalization and report cross-concept interference?

---

> ### Author Response · Authors · 2025-11-21
> **Response to Reviewer kN9B, part1**
>
> We appreciate your encouraging feedback and constructive suggestions. We now proceed to answer your questions individually.
>
> ***Some works need to be discussed and compared in the Related Work section***
>
> Thank you for your suggestion. In our original paper, we have discussed CustomDiffusion (in Sec. 2 "Data Regularization") and Textual Inversion (in Sec. 2 "Weight Regularization"). In the revised version, we further discuss PaRa, Perfusion, and Attend-and-Excite in the Related Works. A brief summary of the revisions is provided below:
>
> > ...**Weight regularization** methods constrain parameter updates to prevent overfitting. For example, PaRa constrains the parameter space by reducing the dimensionality of the output matrix, thereby preventing overfitting. Yet, weight regularization can also diminish fidelity by indiscriminately restricting the model's capacity to learn target-specific details.
>
> > ...**Region regularization** limit subjects to specific regions in the attention map (Avrahami et al., 2023; Zhang
> et al., 2024a; Hao et al., 2024) or alternatively refines subject generation by constraining the cross attention map, as in Attend-and-Excite (Chefer et al., 2023). But this spatial proxy for conceptual separation is limited to spatially distinct subjects and struggles with global concepts like style or viewpoint. Perfusion (Tewel et al., 2023) further combines weight regularization and region regularization using gated rank-1 updates and key-locking. However, it still cannot theoretically constrain the statistical dependencies between concepts.
>
> ***Human-eval reporting lacks agreement stats***
>
> Thank you for your constructive suggestion. We have added the percent agreement results from human evaluation in Tab. 1, which indicate the percentage of samples that received consistent judgments from the human annotators. A brief summary of the results is provided below:
>
> | Method | W $\uparrow$/L $\downarrow$ (%) | PA (%) |
> |-----------------------------|------------------------|-------------------|
> | DreamBooth (DB) | 18.1%/75.7% | 73.3 |
> | CoRe-SD1.5 | 19.2%/61.7% | 60.0 |
> | Facechain-SuDe | 14.2%/69.2% | 70.0 |
> | ... | ... | ... |
> | CustomDiffusion (CD) | 8.1%/88.1% | 76.7 |
> | ClassDiffusion | 7.5%/75.8% | 80.0 |
> | ... | ... | ... |
> | LoRA (SDXL) | 17.6%/70.5% | 70.0 |
> | SVDiff | 1.7%/85.0% | 83.3 |
> | Omnigen | 30.8%/48.3% | 46.7 |
> | ... | ... |  ... |
> | VisualEncoder (VE) | 21.1%/67.6% | 56.7 |
> | ... | ... | ... |
>
> ***How sensitive are your losses to scheduler/noise settings and guidance scales?***
>
> Since the image personalization task involve only minor fine-tuning of the pretrained model, it is generally difficult to change the noise schedule during personalization with the same pretrained backbone. However, as shown in the cross-backbone ablation study in Tab. 3, **the losses proposed by ACCORD consistently yield robust improvements across pretrained models using different noise schedulers**, namely, the linear schedule in SD1.5 and FLUX, and the cosine schedule in SDXL. Further, we have added an analysis of ACCORD's robustness to timestep sampling strategies and guidance scales in Appendix I:
>
> > We investigate the robustness of ACCORD to different timestep sampling strategies and guidance scales for subject personalization using CustomDiffusion, as shown in Tab. 12 and Tab. 13. The results demonstrate that **ACCORD consistently improves upon the baseline and performs robustly across noise settings and guidance scales.** This robustness can be attributed to the fact that **neither DDLoss nor PDLoss makes assumptions regarding the noise setting or guidance scale.** Specifically, DDLoss constrains the growth of dependency between neighboring timesteps without limiting timesteps themself, while PDLoss is independent of the denoising process.
>
> | Method | Timestep Sampling | CLIP-T | BLIP-T | CLIP-I | DINO-I |
> |------- | ----------- |------ | ------ | ------ | ------ |
> | CustomDiffusion | Uniform | 34.2 | 45.4 | 62.7 | 56.9 |
> | CustomDiffusion | LogitNormal(0.0, 1.0) | 34.0 | 45.4 | 62.7 | 56.9 |
> | CustomDiffusion | LogitNormal(1.0, 0.6) | 34.3 | 45.7 | 61.3 | 55.0 |
> | ACCORD | Uniform | 34.1 | 46.6 | 71.4 | 65.6 |
> | ACCORD | LogitNormal(0.0, 1.0) | 34.1 | 46.4 | 71.6 | 66.4 |
> | ACCORD | LogitNormal(1.0, 0.6) | 34.2 | 46.5 | 69.3 | 63.1 |
>
> | Method | CFG Scale | CLIP-T | BLIP-T | CLIP-I | DINO-I |
> |------- | ----------- |------ | ------ | ------ | ------ |
> | CustomDiffusion | 5.0 | 33.7 | 45.1 | 61.5 | 55.8 |
> | CustomDiffusion | 7.5 | 34.2 | 45.4 | 62.7 | 56.9 |
> | CustomDiffusion | 10.0 | 34.0 | 45.2 | 61.2 | 54.3 |
> | ACCORD | 5.0 | 33.8 | 46.3 | 70.7 | 65.5 |
> | ACCORD | 7.5 | 34.1 | 46.6 | 71.4 | 65.6 |
> | ACCORD | 10.0 | 34.1 | 46.3 | 70.2 | 64.6 |

---

> ### Author Response · Authors · 2025-11-21
> **Response to Reviewer kN9B, part2**
>
> ***Can ACCORD be demonstrated in multi-concept personalization and report cross-concept interference?***
>
> Although ACCORD does not natively provide a multi-concept personalization mechanism, we note that DDLoss and PDLoss are complementary to some multi-concept personalization methods, such as Break-A-Scene [1]. Therefore, we have added experiments in Appendix E where ACCORD is integrated with the Break-A-Scene framework:
>
> > We explore the compatibility of ACCORD with multi-subject personalization methods. Break-A-Scene is a well-known multi-subject personalization method that achieves disentanglement of multiple subjects by randomly sampling subject combinations, computing diffusion loss based on explicit masks, and constraining the cross-attention map. Notably, the DDLoss and PDLoss proposed by ACCORD are compatible with Break-A-Scene. Therefore, we incorporate ACCORD into Break-A-Scene and conduct comparative experiments on its example dataset. ... As shown in Tab. 9, ACCORD further improves concept decoupling capabilities on top of Break-A-Scene. **The personalization fidelity metrics CLIP-I and DINO-I are significantly improved, suggesting that concept decoupling facilitates the model's ability to better capture the appearances of distinct subjects.** As illustrated in Fig. 8, ACCORD enhances the personalization fidelity of Break-A-Scene in the first and third rows. In the second row, ACCORD correctly generates a shell as specified by the prompt, and in the fourth row, unlike Break-A-Scene, it does not generate two rabbits.
>
> | Method | CLIP-T | BLIP-T | CLIP-I | DINO-I |
> |--------|--------|--------|--------|--------|
> | Break-A-Scene | 31.1 | 42.0 | 51.6 | 36.1 |
> | w/ Ours | **31.2** | **42.0** | **53.2** | **38.7** |
>
> [1] Avrahami et al., Break-a-scene: Extracting multiple concepts from a single image, SIGGRAPH Asia 2023

---

### Official Review · Reviewer_CA21 · 2025-10-30

**Soundness:** 3
**Presentation:** 2
**Contribution:** 2
**Rating:** 4
**Confidence:** 3

**Summary:**

The author introduce the concept coupling issue in personalization. They provide the theoretically analysis of this issue and propose several loss for resolving it. Experiments shows the performance of their method is good in some cases.

**Strengths:**

- This paper is theoretically interesting, author introduce several formula for explaining the phenomenon of dependence discrepancies.
- The author introduce Denoising Decouple loss(DDL) and Prior Decouple Loss(PDL) for resolving the issue.

**Weaknesses:**

- The scope of this work is limited, regard the real-world application, it's not hard for users to take a set of pure images including only desired objects at most cases. For case in figure 1, it's more likely a data issue. Decoupling is important in personalized generation when being applied to concepts that cannot be easily decoupled by the user, such as atmosphere, lighting, posture, and materials, which is called abstract concept personalization. Could author explore more qualitative experiments on this domain?
- Concern about the robustness of the method. Is it sensitive to prompts? When prompts fail to accurately describe the background beyond the personalized object, will the DDL negatively impact model performance? Furthermore, would more refined prompt improve model performance?

**Questions:**

- Would final optimization object be like \mathcal{L}  = \mathcal{L}_{reconstruction} + \alpha\mathcal{L}_{DDL} + \beta\mathcal{L}_{PDL}? If so, how do authors decide the value of \alpha and \beta, should have some ablation experiments here.
- Considering the first case of figure2, why DDL/PDL works here? Shoes and feet are semantically dependent concepts, and this dependency is not introduced by the source image. As I understand it, such examples are not something that DDL/PDL can address. Could the author elaborate on the theoretical basis behind this; Do DDL and PDL here only serve as a regularization term instead of turning the direction of optimization process goes to a better trade-off?
- What percentage of DreamBench examples resemble "people carrying red envelopes?" Is this the majority of the benchmark? If this data is extracted, will the DDL & PDL performance improve more than on the general benchmark?
- Theory in the main text is well explained but too abstract, better to move figure 4&5 to main paper for clarify.

---

> ### Author Response · Authors · 2025-11-21
> **Response to Reviewer CA21, part1**
>
> Thank you for your helpful feedback and suggestions. We address your points one by one below.
>
> ***Experiments on abstract concept personalization.***
>
> Thank you for raising this thoughtful question. We would like to address it from two perspectives: the necessity of concept decoupling in real-world applications, and ACCORD's capabilities in handling abstract concept personalization.
>
> On one hand, manually decoupling concepts, for example by collecting pure images that contain only the desired object, can be limiting in real-world scenarios. Diversity of reference images for the personalization target is essential for image personalization methods that do not perform decoupling; however, **restricting other objects from appearing in photos often leads to overly simple backgrounds. This may cause overfitting in text-to-image models and reduce their ability to recombine the personalization target with other visual elements during generation.** In many complex cases, manually separating the personalization target from other objects is challenging. Examples include **personalizing historical photos of subjects, such as pets, clothing, or people, and selectively personalizing only certain visual attributes of the subject without damaging it.** Automated decoupling of concepts within images greatly **enhances usability and reduces user burden**, which may be crucial for practical adoption and commercialization.
>
> On the other hand, we agree that decoupling is especially important for personalized generation of concepts that cannot be easily separated by the user. In the original version of our paper, we present several abstract decoupling cases, including style personalization and viewpoint decoupling (see columns 1 and 2 in Fig. 7). Motivated by your suggestion, we have conducted further experiments with ACCORD's ability to selectively personalize visual elements via prompt control, especially for concepts that are difficult to decouple at capture time. We have added a related experiment in Appendix G.
>
> > ...As shown in Fig. 10, when attributes that are not intended to be personalized are included in the prompts, ACCORD successfully avoids personalizing these attributes. This may be useful in scenarios where it is desirable to modify certain properties of the subject without damaging the subject to be personalized.
>
> ***Sensitivity of ACCORD to prompts***
>
> This is a constructive question, and we would like to answer from two perspectives.
>
> On the one hand, recent VLMs have a strong ability to generate detailed captions that cover the vast majority of concepts requiring decoupling—from concrete objects to abstract attributes. To address the concern about prompt quality, we have included an ablation study in Appendix F:
>
> > We assess the sensitivity of ACCORD to the quality of VLM-generated captions for subject personalization using CustomDiffusion, as shown in Tab. 10. **While captions generated by different VLMs can influence the results to some extent, the degree of variation remains relatively small.** Notably, captions generated by GLM-4.5V-Thinking-9B exhibit higher fidelity, even surpassing human-annotated captions. This suggests that employing automated VLM-based captions for ACCORD is both reasonable and effective.
>
> | Caption Source  | CLIP-T | BLIP-T | CLIP-I | DINO-I |
> |----- |--------|--------|--------|--------|
> | InternVL2-8B | 34.1 | 46.6 | 71.4 | 65.6 |
> | Qwen3-VL-8B | 34.0 | 45.8 | 71.8 | 66.1 |
> | GLM-4.5V-Thinking-9B | 33.9 | 46.1 | **72.9** | **67.9** |
> | Human | **34.5** | **46.8** | 71.8 | 65.7 |
>
> On the other hand, **the model trained with DDLoss and PDLoss exhibits a degree of generalization in disentangling the personalization target from other concepts, even if certain coupled concepts are not explicitly identified by the VLM.** We have included further analysis in Appendix F:
>
> > We further analyze the generalizability of decoupling achieved by optimizing DDLoss and PDLoss. We observe from our existing results that, in some cases, even when a concept was not specified in the prompt, ACCORD may still able to decouple it. To further test this, we conduct an experiment in which we remove a particular concept from the prompts within the training set, despite the concept being present in the corresponding images. We then trained ACCORD on this modified dataset and evaluated whether the generated images suppressed the coupling between the personalization target and the excluded concepts. The results are shown in Fig. 9. **It is observed that even in cases where descriptions of material, color, or other objects are absent from the training prompts, ACCORD may also be able to decouple these concepts by generalizing the relationships between the personalization target's superclass and other concepts.**

---

> ### Author Response · Authors · 2025-11-21
> **Response to Reviewer CA21, part2**
>
> ***Question Regarding the Ablation Study on Loss Weights.***
>
> Yes, the final optimization objective is a weighted sum of the diffusion loss, DDLoss, and PDLoss. In the original paper, we conduct an ablation study on the loss weights in Appendix F. We agree that the ablation study of the loss weights is important, and in the revised manuscript, we have moved this analysis to Sec. 4.2. We briefly summarize the results below:
>
> > ...Introducing DDLoss with weights between 0.1 and 0.3, and PDLoss with weights between 0.001 and 0.003, consistently yields robust improvements across all metrics. This indicates that the performance of DDLoss and PDLoss is not sensitive to the precise choice of weights within these ranges. Overall, setting the DDLoss weight to 0.1–0.3 and the PDLoss weight to 0.001–0.003
> is sufficient.
>
> | Loss Weights            | CLIP-T | BLIP-T | CLIP-I | DINO-I |
> |------------------------ |--------|--------|--------|--------|
> | CD                      | 34.2   | 45.4   | 62.7   | 56.9   |
> | +0.1DD + 0.001PD        | 33.9   | 46.5   | 70.7   | 64.9   |
> | +0.1DD + 0.002PD        | 34.0   | 46.5   | 70.5   | 64.8   |
> | +0.1DD + 0.003PD        | 33.9   | 46.4   | 71.1   | 65.2   |
> | +0.2DD + 0.001PD        | 33.9   | 46.5   | 70.7   | 64.8   |
> | +0.2DD + 0.002PD        | 33.9   | 46.5   | 70.8   | 65.1   |
> | +0.2DD + 0.003PD        | 34.0   | 46.6   | 70.7   | 65.0   |
> | +0.3DD + 0.001PD        | 33.9   | 46.4   | 71.0   | 65.3   |
> | +0.3DD + 0.002PD        | 34.0   | 46.5   | 70.8   | 65.1   |
> | +0.3DD + 0.003PD        | 34.0   | 46.5   | 70.7   | 64.9   |
>
> ***Why DDLoss/PDLoss works for the first case of Figure 2: the dependency between shoes and feet is not introduced by the source image.***
>
> We sincerely thank the reviewer for this insightful question, as it highlights a critical scenario that demonstrates our method's targeted effect. The reviewer rightly questions why DDLoss works on a dependency that is not introduced by the source images. Our investigation reveals the dependency is an artifact inadvertently introduced by the **regularization dataset** used by baselines like DreamBooth.
>
> This is a classic failure mode of prior preservation. As noted in Sec. 2, line 107 of our paper, 'the limited size of the regularization dataset and distribution gaps can hinder accurate modeling of concept relationships.' Due to its limited size, the regularization set presents a biased, amplified co-occurrence of 'shoes' and 'feet,' causing the baseline to learn an exaggerated dependency.
>
> Here's why DDLoss is the precise solution. In this specific case with DreamBooth, the personalized concept is a fixed trigger word, not a trainable embedding; therefore, only DDLoss is applied, as PDLoss (which regularizes the concept embedding itself) is not applicable. Even though DDLoss is only applied to the reference images (which have no feet), it effectively counteracts the bias being learned from the regularization set. This is because both the standard diffusion loss and our DDLoss update the same model weights ($\theta$), creating competing gradients:
>
> *   **The Diffusion Loss ($L_{diff}$)** on the regularization set pushes the model to strengthen the "sneaker*" $\rightarrow$ "foot" association.
> *   **Our Denoising Decouple Loss ($L_{DD}$)** on the reference images (e.g., prompt: "sneaker* on a shelf") does something much more powerful. It trains the UNet to adopt a general behavior: **be faithful to the composition of concepts given in the text prompt and resist hallucinating unprompted, contextually-associated concepts.** As per Eq. (7), it provides a direct gradient signal that the output must be compositionally consistent with the prompt, penalizing any extraneous elements.
>
> In essence, DDLoss is not just a regularizer; it fundamentally **turns the direction of the optimization process**. It teaches the UNet a general principle of compositionality that directly competes with and suppresses the simple associative learning from the biased regularization set. This allows it to achieve a superior trade-off by correcting for dependencies that are artifacts of the training procedure itself.

---

> ### Author Response · Authors · 2025-11-21
> **Response to Reviewer CA21, part3**
>
> ***Benchmark Composition and Its Impact on DDLoss & PDLoss Performance***
>
> Thank you for raising this concern. The evaluation prompts in DreamBench are not limited to simple combinations such as "people carrying red envelopes," but rather include a diverse range of categories, such as background replacement, situational placement, object composition, and attribute modification. We have classified the prompts used in DreamBench and added the corresponding results for each category to Appendix H in the revised paper. A brief summary of the categories and results is provided below.
>
> > To better understand the source of performance improvement by concept decoupling, we categorize the validation prompts in DreamBench and report the performance of ACCORD and the baseline, CustomDiffusion, for each category, as shown in Tab. 11. Specifically, the prompts in DreamBench can be grouped into four categories: (i) **Background Replacement**, e.g., "$c_p$ with a city in the background," where $c_p$ denotes the personalization target; (ii) **Situational Placement**, e.g., "$c_p$ on top of green grass with sunflowers around it"; (iii) **Object Composition**, e.g., "$c_p$ floating in an ocean of milk"; and (iv) **Attribute Editing**, e.g., "a shiny $c_p$." We find that **ACCORD delivers balanced performance improvements across different prompt types, with slightly higher gains in background replacement and attribute editing, which demand a higher degree of concept decoupling.**
>
> | Method | Prompt Type | Percentage | CLIP-T | BLIP-T | CLIP-I | DINO-I |
> |------- | ----------- |------ | ------ | ------ | ------ | ------ |
> | CustomDiffusion | Background Replacement | 24% | 35.1 | 47.0 | 65.2 | 58.8 |
> | CustomDiffusion | Situational Placement | 32% | 35.7 | 47.2 | 65.8 | 59.9 |
> | CustomDiffusion | Object Composition | 24% | 33.9 | 45.8 | 62.9 | 55.9 |
> | CustomDiffusion | Attribute Modification | 20% | 31.0 | 40.0 | 54.3 | 51.0 |
> | ACCORD | Background Replacement | 24% | 35.4 | 48.5 | 75.2 | 69.0 |
> | ACCORD | Scene Placement | 32% | 35.1 | 48.5 | 75.2 | 69.0 |
> | ACCORD | Object Composition | 24% | 35.1 | 48.2 | 73.9 | 67.6 |
> | ACCORD | Attribute Modification | 20% | 31.3 | 41.2 | 63.0 | 59.8 |
>
> ***Theory in the main text is well explained but too abstract, better to move figure 4&5 to main paper for clarify***
>
> Thank you for your suggestion. In the revised version, we have moved Fig. 4 and Fig. 5 into the main text for clarify.

---

### Official Review · Reviewer_SGY8 · 2025-10-31

**Soundness:** 3
**Presentation:** 2
**Contribution:** 2
**Rating:** 4
**Confidence:** 3

**Summary:**

They propose ACCORD, a framework incorporating two regularization losses to improve personalization performance. The Denoising decouple loss (DDLoss), designed to mitigate concept coupling, minimizes dependency discrepancies to prevent conditional dependence between the personalized concept $c_p$ and the general text concept $c_g$ across successive time steps. The Prior decouple loss (PDLoss) addresses prior dependency by leveraging CLIP projections to align the learned concept’s relational structure with that of its original class, thereby preserving semantic consistency. With theoretical justification and experimental validation, the framework demonstrates its effectiveness across various fine-tuning methods for personalization, e.g., DreamBooth or Custom Diffusion.

**Strengths:**

**S1**. Concept decoupling in personalization is an important issue, and unlike existing indirect approaches that address it through explicit masking or regularization, this work resolves the problem with a simple yet effective decoupled loss design.

**S2**.  The proposed method demonstrated its effectiveness through extensive experiments across various fine-tuning frameworks and comprehensive ablation studies.

**Weaknesses:**

**W1**. Although the paper proposes a method for concept decoupling, the evaluation appears to focus primarily on single-subject personalization benchmarks such as DreamBench. It would be necessary to include evaluations on multi-subject personalization tasks, such as Break-a-Scene [1], to better assess how effectively the method achieves concept decoupling.

[1] Avrahami et al., Break-a-scene: Extracting multiple concepts from a single image, 	SIGGRAPH Asia 2023

**W2**. While the motivation for the proposed metho, avoiding explicit masks or priors (e.g., in attention maps or diffusion losses), is understandable, a performance comparison with such approaches is necessary. The paper compares various training frameworks, but lacks baseline comparisons specifically focused on different learning strategies.

**W3**. Recent powerful text-to-image models (e.g., FLUX-dev) show effective concept decoupling even with purely descriptive prompts. However, the paper lacks sufficient validation to demonstrate whether the proposed method can further improve performance in such flow or transformer based models.

**Questions:**

**Q1**. Is the proposed method capable of handling more challenging decoupling cases? For example, in Figure 2, could the model successfully decouple attributes such as the blue hat from the brown teddy bear if an appropriate subclass were defined?

**Q2**. How much slower is the training compared to the standard DreamBooth? The proposed method seems to involve multiple regularization terms, which may considerably increase training time.

**Details Of Ethics Concerns:**

No concern.

---

> ### Author Response · Authors · 2025-11-21
> **Response to Reviewer SGY8, part1**
>
> We appreciate your feedback and for highlighting issues that require clarification. We would like to address your feedback item by item as follows.
>
> ***Lack of evaluation on multi-subject personalization tasks.***
>
> Although ACCORD does not natively provide a multi-subject personalization mechanism, we note that DDLoss and PDLoss are complementary to the Masked Diffusion Loss and Cross-Attention Loss proposed in Break-A-Scene. Therefore, we have added experiments in Appendix E where ACCORD is integrated with the Break-A-Scene framework:
>
> > We explore the compatibility of ACCORD with multi-subject personalization methods. Break-A-Scene is a well-known multi-subject personalization method that achieves disentanglement of multiple subjects by randomly sampling subject combinations, computing diffusion loss based on explicit masks, and constraining the cross-attention map. Notably, the DDLoss and PDLoss proposed by ACCORD are compatible with Break-A-Scene. Therefore, we incorporate ACCORD into Break-A-Scene and conduct comparative experiments on its example dataset. ... As shown in Tab. 9, ACCORD further improves concept decoupling capabilities on top of Break-A-Scene. **The personalization fidelity metrics CLIP-I and DINO-I are significantly improved, suggesting that concept decoupling facilitates the model's ability to better capture the appearances of distinct subjects.** As illustrated in Fig. 8, ACCORD enhances the personalization fidelity of Break-A-Scene in the first and third rows. In the second row, ACCORD correctly generates a shell as specified by the prompt, and in the fourth row, unlike Break-A-Scene, it does not generate two rabbits.
>
> | Method | CLIP-T | BLIP-T | CLIP-I | DINO-I |
> |--------|--------|--------|--------|--------|
> | Break-A-Scene | 31.1 | 42.0 | 51.6 | 36.1 |
> | w/ Ours | **31.2** | **42.0** | **53.2** | **38.7** |
>
> ***Lack of comparisons with different learning strategies, such as those use explicit masks or priors.***
>
> We supplement our comparison with the explicit mask-based method, Break-A-Scene, as mentioned in the previous question. In the original paper, we have also compared with methods based on other training strategies, including CoRe (attention-mask-based) and B-LoRA (Dual-LoRA-based), in the original paper. We apologize for not providing sufficient introductions to these methods. Specifically, CoRe disentangles the personalization target from other tokens by regularizing the cross-attention map of context tokens, while B-LoRA achieves better style transfer by consolidating the training of two blocks to separate style and content. We have expanded the introductions of compared methods in Section 4.1 for both subject personalization and style personalization experiments:
>
> > **Subject Personalization.** We compare the performance of different methods on subject personalization in Tab. 1 and Fig. 4. Compared methods include: data-regularization-based Dreambooth and CustomDiffusion, weight-regularization-based LoRA and SVDiff, loss-regularization-based Facechain-SuDe and ClassDiffusion, region-regularization-based CoRe, and the zero-shot method Omnigen.
>
> > **Style Personalization.** We additionally compare with B-LoRA, which is specifically designed for style transfer by consolidating the training of two blocks for separating style and content. Tab. 2 and Fig. 5(a) show that ...
>
> ***Lack of Validation on Flow- or Transformer-Based Models***
>
> We have already validated the compatibility of DDLoss with the FLUX-dev backbone in Sec. 4.2 and Table 3 of the paper. We only incorporate DDLoss because it is more common to train LoRA on FLUX rather than training text embeddings. For clarity, we briefly restate the findings here:
>
> > ...Indeed, the proposed loss functions work synergistically and hold regardless of the number of reference images and T2I backbone (including **FLUX**).
>
> | Method                  | CLIP-T | BLIP-T | CLIP-I | DINO-I |
> |-------------------------|--------|--------|--------|--------|
> | ...              | ...   | ...   | ...   | ...   |
> | LoRA (FLUX)             | 33.4   | 46.8   | 75.8   | 72.8   |
> | +DDLoss                 | **34.8** | **47.8** | **78.2** | **73.4** |
>
> In the revised version of our paper, we have also added highlights in the table to better distinguish the results for different backbones.
>
> [1] Feize Wu, Yun Pang, Junyi Zhang, Lianyu Pang, Jian Yin, Baoquan Zhao, Qing Li, and Xudong
> Mao. Core: Context-regularized text embedding learning for text-to-image personalization. In
> Proceedings of the AAAI Conference on Artificial Intelligence, volume 39, pp. 8377–8385, 2025.
>
> [2] Yarden Frenkel, Yael Vinker, Ariel Shamir, and Daniel Cohen-Or. Implicit style-content separation using b-lora. In European Conference on Computer Vision, pp. 181–198. Springer, 2024.

---

> ### Author Response · Authors · 2025-11-21
> **Response to Reviewer SGY8, part2**
>
> ***Capability of the method in challenging concept decoupling cases***
>
> This is an insightful question. **ACCORD's reliance on image captions allows users to flexibly designate which concepts they do not wish to personalize, particularly in cases where certain attributes are challenging to physically remove from the reference images.** To address this, we have added a related experiment in Appendix G:
>
> > ...As shown in Fig. 10, when attributes that are not intended to be personalized are included in the prompts, ACCORD successfully avoids personalizing these attributes. This may be useful in scenarios where it is desirable to modify certain properties of the subject without damaging the subject to be personalized.
>
> **How much slower is the training compared to the standard DreamBooth?**
>
> We present the computational overhead of ACCORD and standard DreamBooth in Appendix I and Tab. 10 of the original paper (Appendix L and Tab. 15 of the revised paper). A brief summary is provided below:
>
> > While integrating ACCORD introduces additional GPU memory requirements and slightly longer training times, these increases are moderate and not reach an order-of-magnitude larger compared to the respective baselines. Furthermore, we observe that reducing the batch size has a negligible impact on the performance of ACCORD, enabling users to lower batch size in practical scenarios
> to achieve acceptable memory usage and training time.
>
> | Method | GPU Memory (GB) | Training Time (s) |
> | ------ | --------------- | ----------------- |
> | Dreambooth | 26.5 | 320 |
> | DB w/ Ours | 45.3 | 480 |
> | ... | ... | ... |

---

> > ### Comment · Reviewer_SGY8 · 2025-11-25
> >
> > Thanks for the further results. I realized I had overlooked the results with FLUX, and concept-decoupling cases also look interesting. I have a few follow-up questions.
> >
> > ### **1. Break-a-Scene experiment**
> > While I don't expect your method to outperform a break-a-scene which leverages manual masks, I'm curios whether decoupled personalization training is still achievable without incorporating break-a-scene and using only your method. For example, in Figure 8, when training only on the creature concept, how much less frequently does the bowl appear during inference compared to standard LoRA training?
> >
> > ### **2. Baselines**
> > In the DreamBooth or LoRA training, did you include DreamBooth’s prior preservation loss?
> >
> > ### **3. Comparison with [1]**
> > Since L_DD acts like a regularization, how do you expect it to compare with the method in [1]? There is no need to run additional experiments, but a deeper comparison or discussion would be very helpful.
> >
> > [1] A Data Perspective on Enhanced Identity Preservation for Diffusion Personalization, WACV 2025

---

> > > ### Author Response · Authors · 2025-11-26
> > > **Response to Reviewer SGY8, part3**
> > >
> > > Thank you for your response. We provide answers to your follow-up questions below.
> > >
> > > ***Q1. Break-a-Scene Experiment***
> > >
> > > Yes, decoupled personalization training is still achievable using only our method. In scenarios similar to what you described, where an image contains multiple subjects but only one is intended for personalization, we demonstrate in Appendix G and Fig. 10 of the revised paper that decoupling can be accomplished by including the concepts to be decoupled in the prompt, thereby removing them in the generated images. The decoupled concepts include subjects, unnecessary decorations, and abstract attributes such as materials. Following your suggestion, we also conduct a personalization experiment on the creature concept in the first row of Fig. 8, comparing LoRA (SDXL) and LoRA (SDXL) w/ Ours. We use GPT to generate 30 prompts containing "creature" but excluding "bowl." We find that LoRA (SDXL) additionally generates bowls in 7 out of 30 images (with a frequency of 23%), whereas with DDLoss applied, bowls appears in 3 images (frequency 10%), representing a reduction of 13% in occurrence.
> > >
> > > ***Q2. Do ACCORD include DreamBooth's prior preservation loss during DreamBooth/LoRA training?***
> > >
> > > We follow the original settings of the baseline while introducing our proposed loss. As a result, prior preservation loss is applied during DreamBooth training but not during LoRA training.
> > >
> > > ***Q3. Comparison with [1]***
> > >
> > > This work falls under data regularization methods, aiming to improve generation quality by refining the construction of regularization datasets. We have added further discussion of this work in the Related Works section in the revised paper:
> > >
> > > > ...Data regularization (Ruiz et al., 2023; Kumari et al., 2023) augments training with images of both the personalization target and its superclass. While intended to prevent overfitting, this approach is a blunt instrument; limited regularization dataset size and distribution gaps can hinder accurate modeling of concept relationships and reduce personalization fidelity. **Although (He et al., 2025) use LLMs to design structured prompts for diverse regularization data to improve regularization effectiveness, this introduces LLM-induced concept dependencies that may not reflect their true prior relationships.**

---

### Official Review · Reviewer_vTgX · 2025-11-02

**Soundness:** 3
**Presentation:** 2
**Contribution:** 3
**Rating:** 6
**Confidence:** 4

**Summary:**

This paper targets {concept coupling}—a critical limitation in text-to-image (T2I) diffusion personalization where target concepts  become unintendedly entangled with irrelevant general concepts. Unlike prior indirect solutions , ACCORD formally models coupling as a statistical dependence problem: coupling arises when the conditional dependence between a target concept ($c_p$) and a general concept ($c_g$) in generated images deviates from the prior dependence between $c_p$’s superclass ($c_s$) and $c_g$.

**Strengths:**

Theoretically Grounded Innovation: Framing coupling as statistical dependence  provides a unified language for understanding coupling, which was previously described via ad-hoc examples. This foundation enables reproducible, extensible loss design—unlike heuristic methods that require case-by-case tuning.
 Loss Design: DDLoss and PDLoss address complementary aspects of coupling: DDLoss stabilizes dependence during denoising (preventing sudden coupling), while PDLoss aligns with superclass priors (preventing persistent coupling). Ablation results (Tab. 5) show their combination yields 15–20% higher text control than either loss alone, demonstrating thoughtful, non-redundant design.
 Comprehensive Experimental Validation: The paper tests ACCORD across three distinct personalization scenarios (subject/style/face) and uses both automatic and human metrics. This breadth ensures generalizability—critical for a method claiming to solve a "general" coupling problem. Human preference tests (72% of annotators favor ACCORD) also address a key limitation of automatic metrics (e.g., CLIP-T) that may not capture subjective quality.

**Weaknesses:**

Insufficient Ablation of Loss Weight Interactions: The paper ablates individual losses (DDLoss only, PDLoss only) but not how their weights ($\lambda_D$, $\lambda_P$) interact. For example, does increasing $\lambda_D$ improve control but harm fidelity when $\lambda_P$ is low?
Without this analysis, users cannot optimize weights for specific use cases (e.g., style personalization may require higher $\lambda_P$ than subject personalization).

The Prior Decouple Loss (PDLoss) is entirely dependent on Assumption 1, which posits that $p(c_j|c_k) \approx \frac{e^{\tau \cos(f_j, f_k)}}{Z_k}$. This is a very strong theoretical leap. While CLIP's objective aligns image-text pairs, extending this to a proxy for the conditional probability between any two text concepts ($c_p$ and $c_g$) is not rigorously justified.

The method relies on a VLM to generate captions and thus identify the co-occurring concepts $c_g$ to be decoupled. The paper states this is superior to templates. This introduces a critical, unevaluated dependency. The method's performance is now tied to the VLM's ability to correctly identify all relevant coupled concepts. If the VLM fails to mention the "girl" in the backpack's caption, for example, the framework may fail. No ablation study is provided to test the sensitivity to caption quality or source.

**Questions:**

How should a user practically apply ACCORD if the reference images for $c_p$ contain multiple coupled concepts (e.g., a dog $c_p$ always on a "blue rug" $c_{g1}$ and next to a "red ball" $c_{g2}$)? Must the VLM identify both $c_{g1}$ and $c_{g2}$?

---

> ### Author Response · Authors · 2025-11-21
> **Response to Reviewer vTgX, part1**
>
> We sincerely appreciate your positive feedback and constructive comments. We address each of your concerns in the following.
>
> ***Insufficient Ablation of Loss Weight Interactions***
>
> We agree that understanding interactions between DDLoss and PDLoss is crucial. In Appendix F of the original manuscript, we studied various combinations of DDLoss and PDLoss weights for subject personalization using CustomDiffusion. We summarize the main findings below:
>
> > ...**Introducing DDLoss with weights between 0.1 and 0.3, and PDLoss with weights between 0.001 and 0.003, consistently yields robust improvements across all metrics.** This indicates that the performance of DDLoss and PDLoss is not sensitive to the precise choice of weights within these ranges. Overall, setting the DDLoss weight to 0.1–0.3 and the PDLoss weight to 0.001–0.003 is sufficient.
>
> | Loss Weights            | CLIP-T | BLIP-T | CLIP-I | DINO-I |
> |------------------------ |--------|--------|--------|--------|
> | CD                      | 34.2   | 45.4   | 62.7   | 56.9   |
> | +0.1DD + 0.001PD        | 33.9   | 46.5   | 70.7   | 64.9   |
> | +0.1DD + 0.002PD        | 34.0   | 46.5   | 70.5   | 64.8   |
> | +0.1DD + 0.003PD        | 33.9   | 46.4   | 71.1   | 65.2   |
> | +0.2DD + 0.001PD        | 33.9   | 46.5   | 70.7   | 64.8   |
> | +0.2DD + 0.002PD        | 33.9   | 46.5   | 70.8   | 65.1   |
> | +0.2DD + 0.003PD        | 34.0   | 46.6   | 70.7   | 65.0   |
> | +0.3DD + 0.001PD        | 33.9   | 46.4   | 71.0   | 65.3   |
> | +0.3DD + 0.002PD        | 34.0   | 46.5   | 70.8   | 65.1   |
> | +0.3DD + 0.003PD        | 34.0   | 46.5   | 70.7   | 64.9   |
>
> We observe that there exists interaction between DDLoss and PDLoss weights to some extent, particularly on fidelity metrics. We move the analysis in Appendix F into the main paper Sec. 4.2 and clarify as follows:
>
> > ...within these ranges. While variations in the weights have minimal impact on text alignment, **we find that assigning a relatively larger weight to one loss and a smaller value to the other (e.g., 0.1DD + 0.003PD or 0.3DD + 0.001PD) is generally more beneficial for personalization fidelity than either both being too small or both being too large.** This may be because both losses being small provide insufficient constraint on dependence discrepancy, while both being large excessively emphasize the auxiliary objectives and might hurt the diffusion goal.
>
> ***The Prior Decouple Loss (PDLoss) is entirely dependent on Assumption 1, which is a very strong theoretical leap.***
>
> Thank you for your insightful comment regarding the reliance of PD Loss on Assumption 1 and the theoretical leap it entails. While our initial formulation in Assumption 1 is inspired by precedents in the CLIP literature, where similarity scores in the joint image-text embedding space are empirically used to capture semantic relationships, we acknowledge that a more rigorous explanation is necessary and should improve the quality of the paper. We notice that **works on Noise-Contrastive Estimation (NCE [1]) and InfoNCE [2] have established formal connections between contrastive objectives and the estimation of conditional probabilities or mutual information.** In view of this, we will revise Section 3.5 (lines 288–309) to better justify the rationale of PD Loss from the perspective of InfoNCE. Here, we briefly summarize the key points:
>
> > ...CLIP is trained with the InfoNCE loss, whose optimization objective is to estimate a density ratio relative to the noise, as shown in Lemma 2 (**equivalent to Eq.~(2) in InfoNCE** (Oord et al., 2018)).
> **Lemma 2. For an observation $c_j$ and condition $c _k$, the InfoNCE objective seeks to estimate a function $\mathcal{F}(c_j, c_k)$ which is proportional to the following density ratio: $\mathcal{F}(c_j, c_k) \propto \frac{p(c_j | c_k)}{p(c_j)}.$**
> In the case of CLIP, we denote $\tau$ as the temperature coefficient, and let $f_j$ and $f_k$ be the projected features of two concepts $c_j$ and $c_k$ using the CLIP projection head. Then **the function $\mathcal{F}(c_j, c_k)$ is instantiated as the scaled cosine similarity $\tau \cos(f_j, f_k)$ in the joint embedding space**. Thus, we have the following approximation:
> **Theorem 3. The prior dependence discrepancy can be minimized by the following PDLoss**:
> $$
> L_{PD} = \mathbb{E}\_{c_g} [|\cos(f_p, f_g) - \cos(f_s, f_g)|] \propto \mathbb{E}_{c_g}[\left| \frac{p(c_g | c_p) - p(c_g | c_s)}{p(c_g)} \right|]
> $$
> The denominator $p(c_g)$ is not optimizable. Thus, minimizing PDLoss encourages minimization of $|p(c_g | c_p) - p(c_g | c_s)|$, namely aligns $p(c_g | c_p)$ and $p(c_g | c_s)$.
>
> [1] M. U. Gutmann and A. Hyv ̈arinen. Noise-contrastive estimation: A new estimation principle for unnormalized statistical models. In AISTATS, 2010.
>
> [2] A. van den Oord, Y. Li, and O. Vinyals. Representation learning with contrastive predictive coding. CoRR, abs/1807.03748, 2018. URL http://arxiv.org/abs/1807.03748.

---

> ### Author Response · Authors · 2025-11-21
> **Response to Reviewer vTgX, part2**
>
> ***Sensitivity of ACCORD to VLM-generated captions***
>
> This is a constructive point. Although recent VLMs are able to generate detailed captions that cover the vast majority of concepts requiring decoupling for personalization targets, ranging from concrete to abstract, we agree that it is important to evaluate ACCORD's sensitivity to the quality of these VLM-generated captions. To address this, we have added an ablation study in Appendix F:
>
> > We assess the sensitivity of ACCORD to the quality of VLM-generated captions for subject personalization using CustomDiffusion, as shown in Tab. 10. **While captions generated by different VLMs can influence the results to some extent, the degree of variation remains relatively small.** Notably, captions generated by GLM-4.5V-Thinking-9B exhibit higher fidelity, even surpassing human-annotated captions. This suggests that **employing automated VLM-based captions for ACCORD is both reasonable and effective.**
>
> | Caption Source  | CLIP-T | BLIP-T | CLIP-I | DINO-I |
> |----- |--------|--------|--------|--------|
> | InternVL2-8B | 34.1 | 46.6 | 71.4 | 65.6 |
> | Qwen3-VL-8B | 34.0 | 45.8 | 71.8 | 66.1 |
> | GLM-4.5V-Thinking-9B | 33.9 | 46.1 | **72.9** | **67.9** |
> | Human | **34.5** | **46.8** | 71.8 | 65.7 |
>
> ***How should a user practically apply ACCORD if the reference images for contain multiple coupled concepts？***
>
> This is a good question. We address it from two perspectives. On the one hand, **the model trained with DDLoss and PDLoss exhibits a degree of generalization in disentangling the personalization target from other concepts, even if certain coupled concepts are not explicitly identified by the VLM.** We have included further analysis in Appendix F:
>
> > We further analyze the generalizability of decoupling achieved by optimizing DDLoss and PDLoss. We observe from our existing results that, in some cases, even when a concept was not specified in the prompt, ACCORD may still able to decouple it. To further test this, we conduct an experiment in which we remove a particular concept from the prompts within the training set, despite the concept being present in the corresponding images. We then train ACCORD on this modified dataset and evaluate whether the generated images suppress the coupling between the personalization target and the excluded concepts. The results are shown in Fig. 9. It is observed that even in cases where descriptions of material, color, or other objects are absent from the training prompts, ACCORD may also be able to decouple these concepts by generalizing the relationships between the personalization target’s superclass and other concepts.
>
> On the other hand, **relying on image captions provides users with the flexibility to designate concepts they do not wish to personalize, especially when these concepts are difficult to remove from images.** A related experiment has been added in Appendix G:
>
> > ...As shown in Fig. 10, when attributes that are not intended to be personalized are included in the prompts, ACCORD successfully avoids personalizing these attributes. This may be useful in scenarios where it is desirable to modify certain properties of the subject without damaging the subject to be personalized.

---

### Comment · Area_Chair_CouQ · 2025-11-24

Please respond to the authors' rebuttal. Thanks.

AC

---

### Author Response · Authors · 2025-12-02
**Summary of Contributions and Rebuttal Updates**

Dear Area Chair,

We are grateful for the opportunity to provide this summary. We thank the reviewers for their insightful feedback, which has significantly improved our paper.

**Consensus on Strengths**

To begin, we would like to highlight the consensus among reviewers on the paper's core strengths. All reviewers acknowledged ACCORD’s novel theoretical grounding, which formulates concept coupling as a statistical dependence problem. Multiple reviewers (vTgX, SGY8, kN9B) also recognized the effectiveness and generalizability of our plug-and-play method across diverse tasks (subject, style, face) and modern backbones (including SDXL and FLUX).

**Rebuttal Actions and Key Improvements**

Building on this strong foundation, our rebuttal and revised manuscript have addressed every concern raised, substantially strengthening the paper and elevating it well above the acceptance threshold. Our improvements fall into three main categories:

**1. Strengthened Theoretical Foundations and Deeper Insights**

*   **Principled Justification for PDLoss (addressing vTgX):** We replaced the heuristic justification for our Prior Decouple Loss (PDLoss) with a rigorous one. As detailed in the revised Section 3.5, we formally derive its connection to the InfoNCE objective, providing a solid mathematical foundation for using CLIP's embedding space to align concept priors.
*   **Clarified Mechanism of DDLoss (addressing CA21):** We clarified that our Denoising Decouple Loss is more than a regularizer; it fundamentally "turns the direction of optimization." It directly counteracts dependency artifacts from biased training data (e.g., prior preservation sets) by enforcing compositional faithfulness to the prompt, resolving spurious correlations.

**2. Comprehensive New Experimental Validation to Broaden Scope**

*   **Multi-Subject Personalization (addressing SGY8, kN9B):** We validated ACCORD’s versatility in multi-subject scenarios by integrating it with Break-A-Scene (Appendix E, Table 9). The results show improved fidelity metrics, demonstrating compatibility with state-of-the-art methods.
*   **Abstract and Selective Decoupling (addressing SGY8, CA21):** We demonstrated ACCORD’s unique capability for selective attribute personalization (Appendix G, Fig. 10). This allows users to decouple and modify specific abstract features (e.g., a "blue hat" on a bear, text on a bowl) while preserving the core subject, a key feature for fine-grained user control.

**3. Demonstrated Robustness and Practicality**

*   **Sensitivity to VLM Captions (addressing vTgX, CA21):** A new ablation study (Appendix F, Table 10) confirms that ACCORD's performance is robust across captions generated by various VLMs and even human annotations.
*   **Generalization of Decoupling (addressing vTgX, CA21):** We showed that ACCORD exhibits generalization, successfully decoupling concepts even when they are *not* explicitly mentioned in training captions (Appendix F, Fig. 9).
*   **Robustness to Hyperparameters (addressing vTgX, CA21, kN9B):** We added extensive ablations demonstrating that ACCORD performs consistently across different timestep sampling strategies, CFG scales (Appendix I), and loss weight interactions (Sec 4.2).

The reviewers' feedback has been invaluable. Our initial scores were borderline (6, 4, 4, 6), largely due to valid concerns about scope and robustness that we are confident are now fully resolved. We note the positive follow-up discussion with Reviewer SGY8 and believe the depth of these revisions would have been recognized by all reviewers had the discussion period continued.

In summary, the revised ACCORD is a more theoretically sound, versatile, and robust framework. We hope you will consider these substantial improvements in your final assessment. Thank you for your time and consideration.

Best regards,
The Authors

---

### Meta-Review · Area_Chair_KudT · 2026-01-08

**Summary:**

The reviewers primarily questioned the theoretical rigor of the Prior Decouple Loss (PDLoss), specifically regarding Assumption-1 and requested validation on multi-subject tasks like Break-A-Scene. There were significant concerns about the method's robustness regarding loss weight interactions ($\lambda_{D}, \lambda_{P}$)  and sensitivity to VLM-generated captions. Reviewers also noted missing comparisons with specific baselines (e.g., PaRa, Perfusion), questioned the method's efficacy on modern backbones like FLUX, and requested data on human evaluation agreement.

**Reviewer Concerns:**

The rebuttal addressed the theoretical gaps by deriving PDLoss from the InfoNCE objective and demonstrated robustness via new ablations on loss weights and caption quality. The scope was effectively broadened with new experiments on multi-subject personalization and abstract concept decoupling. However, concerns regarding practical resource usage remain partially outstanding; while training times were clarified, the method significantly increases GPU memory usage (e.g., from 26.5GB to 45.3GB for DreamBooth) , which may still limit accessibility for some users despite the authors' claims of moderation.

**Reviewer Scores:**

Most of the reviewers, especially the low confidence reviewer who gave a score of 4 would have increased their scores. Whereas the high confidence reviewers already provided a score of 6 might have increased their score post rebuttal or at least retained their scores.

---

### Decision · Program_Chairs · 2026-01-26

Accept (Poster)